# L-Shapley and C-Shapley: Efficient Model Interpretation for Structured Data

**Jianbo Chen**[*]    **Le Song**[†,§]    **Martin J. Wainwright**[*,‡]    **Michael I. Jordan**[*]
UC Berkeley[*], Georgia Institute of Technology[†], Ant Financial[§], Voleon Group[‡]

## Abstract

Instancewise feature scoring is a method for model interpretation, which yields, for each test instance, a vector of importance scores associated with features. Methods based on the Shapley score have been proposed as a fair way of computing feature attributions, but incur an exponential complexity in the number of features. This combinatorial explosion arises from the definition of Shapley value and prevents these methods from being scalable to large data sets and complex models. We focus on settings in which the data have a graph structure, and the contribution of features to the target variable is well-approximated by a graph-structured factorization. In such settings, we develop two algorithms with linear complexity for instancewise feature importance scoring on black-box models. We establish the relationship of our methods to the Shapley value and a closely related concept known as the Myerson value from cooperative game theory. We demonstrate on both language and image data that our algorithms compare favorably with other methods using both quantitative metrics and human evaluation.

## 1 Introduction

Although many black box machine learning models, such as random forests, deep neural networks, and kernel methods, can produce highly accurate prediction in many applications, such prediction often comes at the cost of interpretability. Ease of interpretation is a crucial criterion when these tools are applied in areas such as medicine, financial markets, and criminal justice; for more background, see the discussion paper by Lipton (2016) as well as references therein.

In this paper, we study instancewise feature importance scoring as a specific approach to the problem of interpreting the predictions of black-box models. Given a predictive model, such a method yields, for each instance to which the model is applied, a vector of importance scores associated with the underlying features. The instancewise property means that this vector, and hence the relative importance of each feature, is allowed to vary across instances. Thus, the importance scores can act as an explanation for the specific instance, indicating which features are the key for the model to make its prediction on that instance.

There is now a large body of research focused on the problem of scoring input features based on the prediction of a given instance (see, e.g., Shrikumar et al., 2017; Bach et al., 2015; Ribeiro et al., 2016; Lundberg & Lee, 2017; Štrumbelj & Kononenko, 2010; Baehrens et al., 2010; Datta et al., 2016; Sundararajan et al., 2017). Of most relevance to this paper is a line of recent work (Štrumbelj & Kononenko, 2010; Lundberg & Lee, 2017; Datta et al., 2016) that has developed methods for model interpretation based on Shapley value (Shapley, 1953) from cooperative game theory. The Shapley value was originally proposed as an axiomatic characterization of a fair distribution of a total surplus from all the players, and can be applied to predictive models, in which case each feature is modeled as a player in the underlying game. While the Shapley value approach is conceptually appealing, it is also computationally challenging: in general, each evaluation of a Shapley value requires an exponential number of model evaluations. Different approaches to circumventing this complexity barrier have been proposed, including those based on Monte Carlo approximation (Štrumbelj & Kononenko, 2010; Datta et al., 2016) and methods based on sampled least-squares with weights (Lundberg & Lee, 2017).

In this paper, we take a complementary point of view, arguing that the problem of explanation is best approached within a model-based paradigm. In this view, explanations are cast in terms of a model,

which may or may not be the same model as used to fit the data. Criteria such as Shapley value, which are intractable to compute when no assumptions are made, can be more effectively computed or approximated within the framework of a model. We focus specifically on settings in which a graph structure is appropriate for describing the relations between features in the data (e.g., chains for sequences and grids for images), and distant features according to the graph have weak interaction during the computation of Shapley values. We propose two methods for instancewise feature importance scoring in this framework, which we term *L-Shapley* and *C-Shapley*; here the abbreviations "L" and "C" refer to "local" and "connected," respectively. By exploiting the underlying graph structure, the number of model evaluations is reduced to linear—as opposed to exponential—in the number of features. We demonstrate the relationship of these measures with a constrained form of Shapley value, and we additionally relate C-Shapley with another solution concept from cooperative game theory, known as the Myerson value (Myerson, 1977). The Myerson value is commonly used in graph-restricted games, under a local additivity assumption of the model on disconnected subsets of features. Finally, we apply our feature scoring methods to several state-of-the-art models for both language and image data, and find that our scoring algorithms compare favorably to several existing sampling-based algorithms for instancewise feature importance scoring.

## 2 BACKGROUND AND PRELIMINARIES

We begin by introducing some background and notation for instancewise feature importance scoring and the Shapley value.

### 2.1 IMPORTANCE OF A FEATURE SUBSET

We are interested in studying models that are trained to perform prediction, taking as input a feature vector $x \in \mathcal{X} \subset \mathbb{R}^d$ and predicting a response or output variable $y \in \mathcal{Y}$. We assume access to the output of a model via a conditional distribution, denoted by $\mathbb{P}_m(\cdot|x)$, that provides the distribution of the response $Y \in \mathcal{Y}$ conditioned on a given vector $X = x$ of inputs. For any given subset $S \subset \{1, 2, \ldots, d\}$, we use $x_S = \{x_j, j \in S\}$ to denote the associated sub-vector of features, and we let $\mathbb{P}_m(Y \mid x_S)$ denote the induced conditional distribution when $\mathbb{P}_m$ is restricted to using only the sub-vector $x_S$. In the corner case in which $S = \emptyset$, we define $\mathbb{P}_m(Y \mid x_\emptyset) := \mathbb{P}_m(Y)$. In terms of this notation, for a given feature vector $x \in \mathcal{X}$, subset $S$ and fitted model distribution $\mathbb{P}_m(Y \mid x)$, we introduce the *importance score*

$$v_x(S) := \mathbb{E}_m \left[ -\log \frac{1}{\mathbb{P}_m(Y \mid x_S)} \,\Big|\, x \right],$$

where $\mathbb{E}_m[\cdot \mid x]$ denotes the expectation over $\mathbb{P}_m(\cdot \mid x)$. The importance score $v_x(S)$ has a coding-theoretic interpretation: it corresponds to the negative of the expected number of bits required to encode the output of the model based on the sub-vector $x_S$. It will be zero when the model makes a deterministic prediction based on $x_S$, and larger when the model returns a distribution closer to uniform over the output space.

There is also an information-theoretic interpretation to this definition of importance scores, as discussed in Chen et al. (2018). In particular, suppose that for a given integer $k < d$, there is a function $x \mapsto S^*(x)$ such that, for all almost all $x$, the $k$-sized subset $S^*(x)$ maximizes $v_x(S)$ over all subsets of size $k$; then we are guaranteed that the mutual information $I(X_{S^*(X)}, Y)$ between $X_{S^*(X)}$ and $Y$ is maximized, over any conditional distribution that generates a subset of size $k$ given $X$. The converse is also true.

In many cases, class-specific importance is favored, where one is interested in seeing how important a feature subset $S$ is to the predicted class, instead of the prediction as a conditional distribution. In order to handle such cases, it is convenient to introduce the degenerate conditional distribution

$$\hat{\mathbb{P}}_m(y \mid x) := \begin{cases} 1 \text{ if } y \in \arg\max_{y'} \mathbb{P}_m(y' \mid x), \\ 0 \text{ otherwise.} \end{cases}$$

We can then define the importance of a subset $S$ with respect to $\hat{\mathbb{P}}_m$ using the modified score

$$v_x(S) := \hat{\mathbb{E}}_m \left[ -\log \frac{1}{\mathbb{P}_m(Y \mid x_S)} \,\Big|\, x \right],$$

which is the expected log probability of the predicted class given the features in $S$.

**Estimating the conditional distribution:** In practice, we need to estimate—for any given feature vector $\bar{x} \in \mathcal{X}$—the conditional probability functions $\mathbb{P}_m(y \mid \bar{x}_S)$ based on observed data. Past work has used one of two approaches: either estimation based on empirical averages (Štrumbelj & Kononenko, 2010), or plug-in estimation using a reference point (Datta et al., 2016; Lundberg & Lee, 2017).

*Empirical average estimation*: In this approach, we first draw a set of feature vector $\{x^j\}_{j=1}^M$ by sampling with replacement from the full data set. For each sample $x^j$, we define a new vector $\tilde{x}^j \in \mathbb{R}^d$ with components $(\tilde{x}^j)_i$ equal to $x_i^j$ if $i \in S$ and $\bar{x}_i$ otherwise. Taking the empirical mean of $\mathbb{P}_m(y \mid \tilde{x}^j)$ over $\{\tilde{x}^j\}$ then provides an estimate of $\mathbb{P}_m(y \mid \bar{x}_S)$.

*Plug-in estimation*: In this approach, the first step is to specify a reference vector $x^0 \in \mathbb{R}^d$ is specified. We then define the vector $\tilde{x} \in \mathbb{R}^d$ with components $(\tilde{x})_i$ equal to $x_i$ if $i \in S$ and $x_i^0$ otherwise. Finally, we use the conditional probability $\mathbb{P}_m(y \mid \tilde{x})$ as an approximation to $\mathbb{P}_m(y \mid \bar{x}_S)$. The plug-in estimate is more computationally efficient than the empirical average estimator, and works well when there exist appropriate choices of reference points. We use this method for our experiments, where we use the index of padding for language data, and the average pixel strength of an image for vision data.

## 2.2 SHAPLEY VALUE FOR MEASURING INTERACTION BETWEEN FEATURES

Consider the problem of quantifying the importance of a given feature index $i$ for feature vector $x$. A naive way of doing so would be by computing the importance score $v_x(\{i\})$ of feature $i$ on its own. However, doing so ignores interactions between features, which are likely to be very important in applications. As a simple example, suppose that we were interested in performing sentiment analysis on the following sentence:

$$\textit{It is not heartwarming or entertaining. It just sucks.} \tag{$\star$}$$

This sentence is contained in a movie review from the IMDB movie data set (Maas et al., 2011), and it is classified as negative sentiment by a machine learning model to be discussed in the sequel. Now suppose we wish to quantify the importance of feature "*not*" in prediction. The word "*not*" plays an important role in the overall sentence as being classified as negative, and thus should be attributed a significant weight. However, viewed in isolation, the word "*not*" has neither negative nor positive sentiment, so that one would expect that $v_x(\{\text{"}not\text{"}\}) \approx 0$.

Thus, it is essential to consider the interaction of a given feature $i$ with other features. For a given subset $S$ containing $i$, a natural way in which to assess how $i$ interacts with the other features in $S$ is by computing the difference between the importance of all features in $S$, with and without $i$. This difference is called the *marginal contribution* of $i$ to $S$, and given by

$$m_x(S, i) := v_x(S) - v_x(S \setminus \{i\}). \tag{1}$$

In order to obtain a simple scalar measure for feature $i$, we need to aggregate these marginal contributions over all subsets that contain $i$. The *Shapley value* (Shapley, 1953) is one principled way of doing so. For each integer $k = 1, \ldots, d$, we let $\mathcal{S}_k(i)$ denote the set of $k$-sized subsets that contain $i$. The Shapley value is obtained by averaging the marginal contributions, first over the set $\mathcal{S}_k(i)$ for a fixed $k$, and then over all possible choices of set size $k$:

$$\phi_x(\mathbb{P}_m, i) := \frac{1}{d} \sum_{k=1}^d \frac{1}{\binom{d-1}{k-1}} \sum_{S \in \mathcal{S}_k(i)} m_x(S, i). \tag{2}$$

Since the model $\mathbb{P}_m$ remains fixed throughout our analysis, we frequently omit the dependence of $\phi_x$ on $\mathbb{P}_m$, instead adopting the more compact notation $\phi_x(i)$.

The concept of Shapley value was first introduced in cooperative game theory (Shapley, 1953), and it has been used in a line of recent work on instancewise feature importance ranking (Štrumbelj & Kononenko, 2010; Datta et al., 2016; Lundberg & Lee, 2017). It can be justified on an axiomatic basis (Shapley, 1953; Young, 1985) as being the unique function from a collection of $2^d$ numbers (one for each subset $S$) to a collection of $d$ numbers (one for each feature $i$) with the following properties: (i) [Additivity] The sum of the Shapley values $\sum_{i=1}^d \phi_x(i)$ is equal to the difference $v_x(\{1, \ldots, d\}) - v_x(\emptyset)$. (ii) [Equal contributions] If $v_x(S \cup \{i\}) = v_x(S \cup \{j\})$ for all subsets $S$, then $\phi_x(i) = \phi_x(j)$. (iii) [Monotonicity] Given two models $\mathbb{P}_m$ and $\mathbb{P}_m$, let $m_x$ and $m_x'$ denote the

associated marginal contribution functions, and let $\phi_x$ and $\phi'_x$ denote the associated Shapley values. If $m_x(S, i) \geq m'_x(S, i)$ for all subsets $S$, then we are guaranteed that $\phi_x(i) \geq \phi'_x(i)$. Note that all three of these axioms are reasonable in our feature selection context.

## 2.3 THE CHALLENGE WITH COMPUTING SHAPLEY VALUES

The exact computation of the Shapley value $\phi_x(i)$ takes into account the interaction of feature $i$ with all $2^{d-1}$ subsets that contain $i$, thereby leading to computational difficulties. Various approximation methods have been developed with the goal of reducing complexity. For example, Štrumbelj & Kononenko (2010) proposed to estimate the Shapley values via a Monte Carlo approximation built on an alternative permutation-based definition of the Shapley value. Lundberg & Lee (2017) proposed to evaluate the model over randomly sampled subsets and use a weighted linear regression to approximate the Shapley values based on the collected model evaluations.

In practice, such sampling-based approximations may suffer from high variance when the number of samples to be collected per instance is limited. (See Appendix E for an empirical evaluation.) For large-scale predictive models, the number of features is often relatively large, meaning that the number of samples required to obtain stable estimates can be prohibitively large. The main contribution of this paper is to address this challenge in a model-based paradigm, where the contribution of features to the response variable respects the structure of an underlying graph. In this setting, we propose efficient algorithms and provide bounds on the quality of the resulting approximation. As we discuss in more detail later, our approach should be viewed as complementary to sampling-based or regresssion-based approximations of the Shapley value. In particular, these methods can be combined with the approach of this paper so as to speed up the computation of the L-Shapley and C-Shapley values that we propose.

## 3 METHODS

In many applications, the features can be associated with the nodes of a graph, and we can define distances between pairs of features based on the graph structure. Intuitively, features distant in the graph have weak interactions with each other, and hence excluding those features in the computation of Shapley value has little effect. For instance, each feature vector $x$ in sequence data (such as language, music etc.), can be associated with a line graph, where positions too far apart in a sequence may not affect each other in Shapley value computation; similarly, each image data is naturally modeled with a grid graph, such that pixels that are far apart may have little effect on each other in the computation of Shapley value.

In this section, we propose modified forms of the Shapley values, referred to as L-Shapley and C-Shapley values, that can be computed more efficiently than the Shapley value by excluding those weak interactions in the structured data. We also show that under certain probabilistic assumptions on the marginal distribution over the features, these quantities yield good approximations to the original Shapley values.

More precisely, given feature vectors $x \in \mathbb{R}^d$, we let $G = (V, E)$ denote a connected graph with nodes $V$ and edges $E \subset V \times V$, where each feature $i$ is associated with a a node $i \in V$, and edges represent interactions between features. The graph induces a distance function on $V \times V$, given by

$$d_G(\ell, m) = \text{number of edges in shortest path joining } \ell \text{ to } m. \qquad (3)$$

In the line graph, this graph distance corresponds to the number of edges in the unique path joining them, whereas it corresponds to the Manhattan distance in the grid graph. For a given node $i \in V$, its *k-neighborhood* is the set

$$\mathcal{N}_k(i) := \{j \in V \mid d_G(i, j) \leq k\} \qquad (4)$$

of all nodes at graph distance at most $k$. See Figure 1 for an illustration for the 2D grid graph.

We propose two algorithms for approximating Shapley value in which features that are either far apart on the graph or features that are not directly connected have an accordingly weaker interaction.

## 3.1 LOCAL SHAPLEY

In order to motivate our first graph-structured Shapley score, let us take a deeper look at Example (⋆). In order to compute the importance score of "*not,*" the most important words to be included are "*heartwarming*" and "*entertaining.*" Intuitively, the words distant from them have a weaker influence

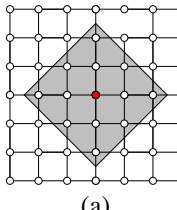 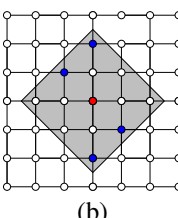 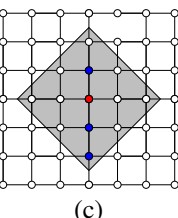

(a)  (b)  (c)

Figure 1: In all cases, the red node denotes the target feature $i$. (a) Illustration of the $k = 2$ graph neighborhood $\mathcal{N}_2(i)$ on the grid graph. All nodes within the shaded gray triangle lie within the neighborhood $\mathcal{N}_2(i)$. (b) A disconnected subset of $\mathcal{N}_2(i)$ that is summed over in L-Shapley but not C-Shapley. (c) A connected subset of $\mathcal{N}_2(i)$ that is summed over in both L-Shapley and C-Shapley.

on the importance of a given word in a document, and therefore have relatively less effect on the Shapley score. Accordingly, as one approximation, we propose the L-Shapley score, which only perturbs the neighboring features of a given feature when evaluating its importance:

**Definition 1.** *Given a model* $\mathbb{P}_m$*, a sample* $x$ *and a feature* $i$*, the* L-Shapley *estimate of order* $k$ *on a graph* $G$ *is given by*

$$\hat{\phi}_x^k(i) := \frac{1}{|\mathcal{N}_k(i)|} \sum_{\substack{T \ni i \\ T \subseteq \mathcal{N}_k(i)}} \frac{1}{\binom{|\mathcal{N}_k(i)|-1}{|T|-1}} m_x(T, i). \tag{5}$$

The coefficients in front of the marginal contributions of feature $i$ are chosen to match the coefficients in the definition of the Shapley value restricted to the neighborhood $\mathcal{N}_k(i)$. We show in Section 4 that this choice controls the error under certain probabilistic assumptions. In practice, the choice of the integer $k$ is dictated by computational considerations. By the definition of $k$-neighborhoods, evaluating all $d$ L-Shapley scores on a line graph requires $2^{2k}d$ model evaluations. (In particular, computing each feature takes $2^{2k+1}$ model evaluations, half of which overlap with those of its preceding feature.) A similar calculation shows that computing all $d$ L-Shapley scores on a grid graph requires $2^{4k^2}d$ function evaluations.

### 3.2 CONNECTED SHAPLEY

We also propose a second algorithm, C-Shapley, that further reduces the complexity of approximating the Shapley value. Coming back to Example $(\star)$ where we evaluate the importance of "*not,*" both the L-Shapley estimate of order larger than two and the exact Shapley value estimate would evaluate the model on the word subset "*It not heartwarming,*" which rarely appears in real data and may not make sense to a human or a model trained on real-world data. The marginal contribution of "*not*" relative to "*It not heartwarming*" may be well approximated by the marginal contribution of "not" to "*not heartwarming.*" This motivates us to proprose *C-Shapley*:

**Definition 2.** *Given a model* $\mathbb{P}_m$*, a sample* $x$ *and a feature* $i$*, the* C-Shapley *estimate of order* $k$ *on a graph* $G$ *is given by*

$$\tilde{\phi}_x^k(i) := \sum_{U \in \mathcal{C}_k(i)} \frac{2}{(|U|+2)(|U|+1)|U|} m_x(U, i), \tag{6}$$

*where* $\mathcal{C}_k(i)$ *denotes the set of all subsets of* $\mathcal{N}_k(i)$ *that contain node* $i$*, and are connected in* $G$*.*

The coefficients in front of the marginal contributions are a result of using Myerson value to characterize a new coalitional game over the graph $G$, in which the influence of disconnected subsets of features are additive. The error between C-Shapley and the Shapley value can also be controlled under certain statistical assumptions. See Section 4 for details.

For text data, C-Shapley is equivalent to only evaluating n-grams in a neighborhood of the word to be explained. By the definition of $k$-neighborhoods, evaluating the C-Shapley scores for all $d$ features takes $\mathcal{O}(k^2 d)$ model evaluations on a line graph, as each feature takes $\mathcal{O}(k^2)$ model evaluations.

## 4 PROPERTIES

In this section, we study some basic properties of the L-Shapley and C-Shapley values. In particular, under certain probabilistic assumptions on the features, we show that they provide good

approximations to the original Shapley values. We also show their relationship to another concept from cooperative game theory, namely that of Myerson values, when the model satisfies certain local additivity assumptions.

## 4.1 Approximation of Shapley value

In order to characterize the relationship between L-Shapley and the Shapley value in terms of some conditional independence assumption between features, we introduce *absolute mutual information* as a measure of dependence. Given two random variables $X$ and $Y$, the absolute mutual information $I_a(X; Y)$ between $X$ and $Y$ is defined as

$$I_a(X; Y) = \mathbb{E}\left[\left|\log \frac{P(X, Y)}{P(X)P(Y)}\right|\right],\tag{7}$$

where the expectation is taken jointly over $X, Y$. Based on the definition of independence, we have $I_a(X; Y) = 0$ if and only if $X \perp\!\!\!\perp Y$. Recall the mutual information (Cover & Thomas, 2012) is defined as $I(X; Y) = \mathbb{E}[\log \frac{P(X, Y)}{P(X)P(Y)}]$. The new measure is more stringent than the mutual information in the sense that $I(X; Y) \leq I_a(X; Y)$. The absolute conditional mutual information can be defined in an analogous way. Given three random variables $X, Y$ and $Z$, we define the absolute conditional mutual information to be $I_a(X; Y \mid Z) = \mathbb{E}[|\log \frac{P(X, Y|Z)}{P(X|Z)P(Y|Z)}|]$, where the expectation is taken jointly over $X, Y, Z$. Recall that $I_a(X; Y \mid Z)$ is zero if and only if $X \perp\!\!\!\perp Y|Z$.

Theorem 1 and Theorem 2 show that L-Shapley and C-Shapley values, respectively, are related to the Shapley value whenever the model obeys a Markovian structure that is encoded by the graph. We leave their proofs to Appendix B.

**Theorem 1.** *Suppose there exists a feature subset $S \subset \mathcal{N}_k(i)$ with $i \in S$, such that*

$$\sup_{U \subset S\setminus\{i\}, V \subset [d]\setminus S} I_a(X_i; X_V | X_U, Y) \leq \varepsilon; \quad \sup_{U \subset S\setminus\{i\}, V \subset [d]\setminus S} I_a(X_i; X_V | X_U) \leq \varepsilon,\tag{8}$$

*where we identify $I_a(X_i; X_V | X_\emptyset)$ with $I_a(X_i; X_V)$ for notational convenience. Then the expected error between the L-Shapley estimate $\hat{\phi}_X^k(i)$ and the true Shapley-value-based importance score $\phi_i(\mathbb{P}_m, x)$ is bounded by $4\varepsilon$:*

$$\mathbb{E}_X |\hat{\phi}_X^k(i) - \phi_X(i)| \leq 4\varepsilon.\tag{9}$$

*In particular, we have $\hat{\phi}_X^k(i) = \phi_X(i)$ almost surely if we have $X_i \perp\!\!\!\perp X_{[d]\setminus S}|X_T$ and $X_i \perp\!\!\!\perp X_{[d]\setminus S}|X_T, Y$ for any $T \subset S \setminus \{i\}$.*

**Theorem 2.** *Suppose there exists a neighborhood $S \subset \mathcal{N}_k(i)$ of $i$, with $i \in S$, such that Condition 8 is satisfied. Moreover, for any connected subset $U \subset S$ with $i \in U$, we have*

$$\sup_{V \subset R(U)} I_a(X_i; X_V | X_{U\setminus\{i\}}, Y) \leq \varepsilon; \quad \sup_{V \subset R(U)} I_a(X_i; X_V | X_{U\setminus\{i\}}) \leq \varepsilon,\tag{10}$$

*where $R(U) := \{i \in [d] - U : \text{ for any } j \in U, (i, j) \notin E\}$. Then the expected error between the C-Shapley estimate $\tilde{\phi}_X^k(i)$ and the true Shapley-value-based importance score $\phi_i(\mathbb{P}_m, x)$ is bounded by $6\varepsilon$:*

$$\mathbb{E}_X |\tilde{\phi}_X^k(i) - \phi_X(i)| \leq 6\varepsilon.\tag{11}$$

*In particular, we have $\hat{\phi}_X^d(i) = \phi_X(i)$ almost surely if we have $X_i \perp\!\!\!\perp X_{R(U)}|X_{U\setminus\{i\}}$ and $X_i \perp\!\!\!\perp X_{R(U)}|X_{U\setminus\{i\}}, Y$ for any $U \subset [d]$.*

## 4.2 Relating the C-Shapley value to the Myerson value

Let us now discuss how the C-Shapley value can be related to the Myerson value, which was introduced by Myerson (1977) as an approach for characterizing a coalitional game over a graph $G$. Given a subset of nodes $S$ in the graph $G$, let $\mathcal{C}_G(S)$ denote the set of connected components of $S$. Thus, if $S$ is a connected subset of $G$, then $\mathcal{C}_G(S)$ consists only of $S$; otherwise, it contains a collection of subsets whose disjoint union is equal to $S$.

Consider a score function $T \mapsto v(T)$ that satisfies the following decomposability condition: for any subset of nodes $S$, the score $v(S)$ is equal to the sum of the scores over the connected components

of $S$:

$$v(S) = \sum_{T \in \mathcal{C}_G(S)} v(T). \tag{12}$$

For any such score function, we can define the associated Shapley value, and it is known as the *Myerson value* on $G$ with respect to $v$. Myerson (1977) showed that the Myerson value is the unique quantity that satisfies both the decomposability property, as well as the properties additivity, equal contributions and monotonicity given in Section 2.2.

In our setting, if we use a plug-in estimate for conditional probability, the decomposability condition (12) is equivalent to assuming that the influence of disconnected subsets of features are additive at sample $x$, and C-Shapley of order $k = d$ is exactly the Myerson value over $G$. In fact, if we partition each subset $S$ into connected components, as in the definition of Myerson value, and sum up the coefficients (using Lemma 1 in Appendix B), then the Myerson value is equivalent to equation 6.

### 4.3 CONNECTIONS WITH RELATED WORK

Let us how methods useful for approximating the Shapley value can be used to speed up the evaluation of approximate L-Shapley and C-Shapley values.

**Sampling-based methods** An alternative definition of the Shapley value defines the contribution of a feature $i$ as the average of the marginal contribution of $i$ to its preceding features over the set of all permutations of $d$ features. Based on this definition, Štrumbelj & Kononenko (2010) propose a Monte Carlo approximation, based on randomly sampling permutations. While L-Shapley is deterministic in nature, it is possible to combine it with this and other sampling-based methods. For example, if one hopes to consider the interaction of features in a large neighborhood $\mathcal{N}_k(i)$ with a feature $i$, where exponential complexity in $k$ becomes a barrier, sampling based on random permutation of local features may be used to alleviate the computational burden.

**Regression-based methods** Lundberg & Lee (2017) proposed to sample feature subsets based on a weighted kernel, and carry out a weighted linear regression to estimate the Shapley value. Strong empirical results were provided using the regression-based approximation, referred to as KernelSHAP; see, in particular, Section 5.1 and Figure 3 of their paper. We can combine such a regression-based approximation with our modified Shapley values to further reduce the evaluation complexity of the C-Shapley values. In particular, for a chain graph, we evaluate the score function over all connected subsequences of length $\leq k$; similarly, on a grid graph, we evaluate it over all connected squares of size $\leq k \times k$.

## 5 EXPERIMENTS

We evaluate the performance of L-Shapley and C-Shapley on real-world data sets involving text and image classification. We compare L-Shapley and C-Shapley with several competitive algorithms for instancewise feature importance scoring on black-box models, including the regression-based approximation known as KernelSHAP (Lundberg & Lee, 2017), SampleShapley (Štrumbelj & Kononenko, 2010), and the LIME method (Ribeiro et al., 2016). We emphasize that our focus is model-agnostic interpretation, and we omit the comparison with methods requiring additional assumptions or specific to a certain class models (e.g., (Sundararajan et al., 2017; Shrikumar et al., 2017; Bach et al., 2015; Karpathy et al., 2015; Strobelt et al., 2018; Murdoch & Szlam, 2017)). For all methods, we choose the objective to be the log probability of the predicted class, and use the plug-in estimate of conditional probability across all methods (see Section 2.1). See Appendix C and D for more experiments on a direct evaluation of the correlation with the Shapley value, and an analysis of sensitivity.

### 5.1 TEXT CLASSIFICATION

Text classification is a classical problem in natural language processing, in which text documents are assigned to predefined categories. We study the performance of L-Shapley and C-Shapley on three popular neural models for text classification: word-based CNNs (Kim, 2014), character-based CNNs (Zhang et al., 2015), and long-short term memory (LSTM) recurrent neural networks (Hochreiter & Schmidhuber, 1997), with the following three data sets on different scales: (i) **IMDB Review with Word-CNN**: A simple word-based CNN model is used on the IMDB movie review data set, achieving a test accuracy of $90.1\%$; (ii) **AG news with Char-CNN**: We implement a character-based CNN on the AG news corpus Zhang et al. (2015), achieving a test accuracy of

| Data Set | Classes | Train Samples | Test Samples | Average #w | Model | Parameters | Accuracy |
|---|---|---|---|---|---|---|---|
| IMDB Review (Maas et al., 2011) | 2 | 25,000 | 25,000 | 325.6 | WordCNN | 351,002 | 90.1% |
| AG's News (Zhang et al., 2015) | 4 | 120,000 | 7,600 | 43.3 | CharCNN | 11,337,988 | 90.09% |
| Yahoo! Answers (Zhang et al., 2015) | 10 | 1,400,000 | 60,000 | 108.4 | LSTM | 7,146,166 | 70.84% |

Table 1: A summary of data sets and models in three experiments. "Average #w" is the average number of words per sentence. "Accuracy" is the model accuracy on test samples.

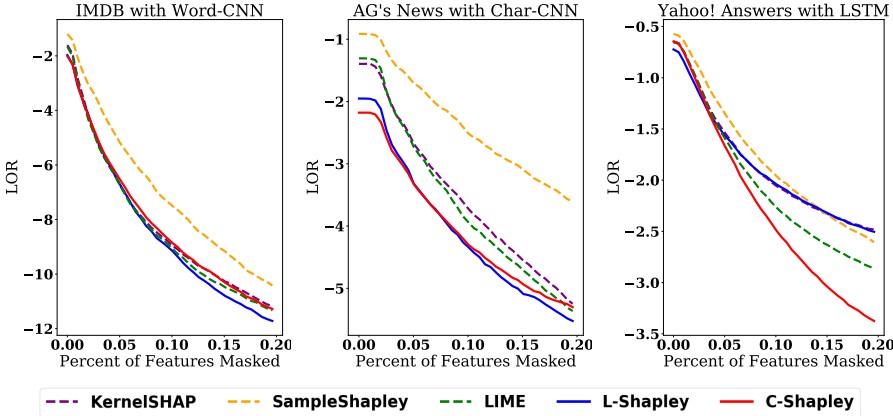

Figure 2: The above plots show the change in log odds ratio of the predicted class as a function of the percent of masked features, on the three text data sets. Lower log odds ratios are better.

| Method | Explanation |
|---|---|
| Shapley | It is not heartwarming or entertaining . It just sucks . |
| C-Shapley | It is not heartwarming or entertaining . It just sucks . |
| L-Shapley | It is not heartwarming or entertaining . It just sucks . |
| KernelSHAP | It is not heartwarming or entertaining . It just sucks . |
| SampleShapley | It is not heartwarming or entertaining . It just sucks . |

Table 2: Each word is highlighted with the RGB color as a linear function of its importance score. The background colors of words with positive and negative scores are linearly interpolated between blue and white, red and white respectively.

90.09%; (iii) **Yahoo! Answers with LSTM**: We train a bidirectional LSTM on the Yahoo! Answers Topic Classification Dataset (Zhang et al., 2015), which achieves a test accuracy of 70.84%. See Table 1 for a summary, and Appendix A for all of the details.

We choose zero paddings as the reference point for all methods, and make $4 \times d$ model evaluations, where $d$ is the number of words for each input. Given the average length of each input (see Table 1), this choice controls the number of model evaluations under $1,000$, taking less than one second in TensorFlow on a Tesla K80 GPU for all the three models. For L-Shapley, we are able to consider the interaction of each word $i$ with the two neighboring words in $\mathcal{N}_1(i)$ given the budget. For C-Shapley, the budget allows the regression-based version to evaluate all $n$-grams with $n \le 4$.

The change in log-odds scores before and after masking the top features ranked by importance scores is used as a metric for evaluating performance, where masked words are replaced by zero paddings. This metric has been used in previous literature in model interpretation (Shrikumar et al., 2017; Lundberg & Lee, 2017). We study how the average log-odds score of the predicted class decreases as the percentage of masked features over the total number of features increases on $1,000$ samples from the test set. Results are plotted in Figure 2.

On IMDB with Word-CNN, the simplest model among the three, L-Shapley, achieves the best performance while LIME, KernelSHAP and C-Shapley achieve slightly worse performance. On AG's news with Char-CNN, L-Shapley and C-Shapley both outperform other algorithms. On Yahoo! Answers with LSTM, C-Shapley outperforms the rest of the algorithms by a large margin, followed by LIME. L-Shapley with order 1, SampleShapley, and KernelSHAP do not perform well for LSTM model, probably because some of the signals captured by LSTM are relatively long $n$-grams.

We also visualize the importance scores produced by different Shapley-based methods on Example ($\star$), which is part of a negative movie review taken from IMDB. The result is shown in Table 2. More visualizations by our methods can be found in Appendix H and Appendix I.

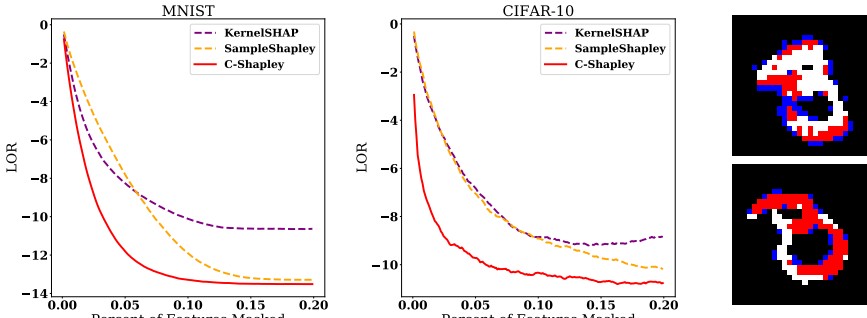

Figure 3: Left and Middle: change in log-odds ratio vs. the percent of pixels masked on MNIST and CIFAR10. Right: top pixels ranked by C-Shapley for a "3" and an "8" misclassified into "8" and "3" respectively. The masked pixels are colored with red if activated (white) and blue otherwise.

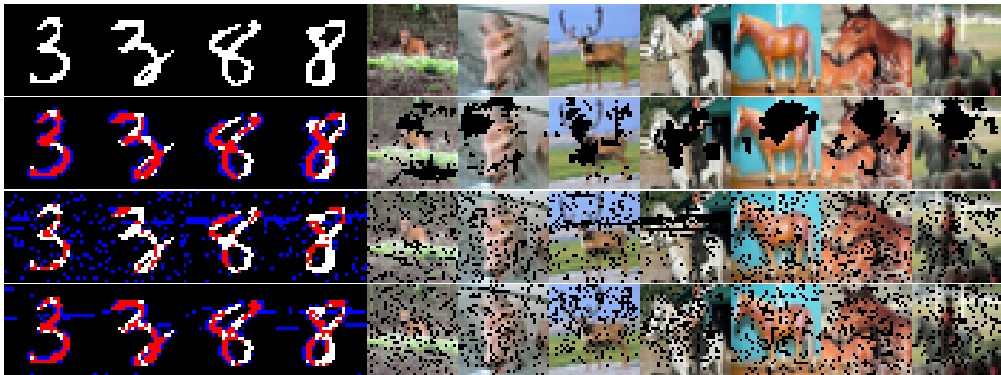

Figure 4: Some examples of explanations obtained for the MNIST and CIFAR10 data sets. The first row corresponds to the original images, with the rows below showing images masked based on scores produced by C-Shapley, KernelSHAP and SampleShapley respectively. For MNIST, the masked pixels are colored with red if activated (white) and blue otherwise.

## 5.2 IMAGE CLASSIFICATION

We carry out experiments in image classification on the MNIST and CIFAR10 data sets: (i) **MNIST**: A subset of MNIST data set (LeCun et al., 1998) composed of digits 3 and 8 is used for better visualization, on which a simple CNN model achieves 99.7% test accuracy; (ii) **CIFAR10**: A subset of the CIFAR10 (Krizhevsky, 2009) composed of deers and horses is used. A convolutional neural network modified from AlexNet (Krizhevsky et al., 2012) achieves 96.1% test accuracy.

We take each pixel as a single feature for both MNIST and CIFAR10. We choose the average pixel strength and the black pixel strength respectively as the reference point for all methods, and make $4 \times d$ model evaluations, where $d$ is the number of pixels for each input image, which keeps the number of model evaluations under $4,000$.

LIME and L-Shapley are not used for comparison because LIME takes "superpixels" instead of raw pixels segmented by segmentation algorithms as single features, and L-Shapley requires nearly sixteen thousand model evaluations when applied to raw pixels.[1] For C-Shapley, the budget allows the regression-based version to evaluate all $n \times n$ image patches with $n \leq 4$.

Figure 3 shows the decrease in log-odds scores before and after masking the top pixels ranked by importance scores as the percentage of masked pixels over the total number of pixels increases on $1,000$ test samples on MNIST and CIFAR10 data sets. C-Shapley consistently outperforms other methods on both data sets. Figure 3 also shows two misclassified digits by the CNN model. Interestingly, the top pixels chosen by C-Shapley visualize the "reasoning" of the model: the important pixels to the model are exactly those which could form a digit from the opposite class.

---

[1]L-Shapley becomes practical if we take small patches of images instead of pixels as single features.

| Algorithm | Modification | Consistency | Standard Deviation | Abs. Score | Words Masked |
|---|---|---|---|---|---|
| Raw | None | 0.880 | 0.960 | 0.811 | N/A |
| L-Shapley | Selected | 0.970 | 0.891 | 1.118 | N/A |
| | Masked | **0.615** | **1.077** | **0.474** | **14.36%** |
| C-Shapley | Selected | **0.990** | **0.500** | **1.441** | N/A |
| | Masked | 0.830 | 0.778 | 0.743 | 14.75% |
| KernelSHAP | Selected | 0.960 | 0.627 | 1.036 | N/A |
| | Masked | 0.660 | 0.818 | 0.492 | 31.60% |

Table 3: Results of human evaluation. "Selected" and "Masked" indicate selected words and masked reviews respectively. Results are averaged over 200 samples. (The best numbers are highlighted.)

Figure 4 provides additional visualization of the results. By masking the top pixels ranked by various methods, we find that the pixels picked by C-Shapley concentrate around and inside the digits in MNIST. For SampleShapley and KernelSHAP, unactivated pixels in MNIST are attributed nonzero scores when evaluated jointly with activated pixels. While one could use post-processing by not choosing unactivated pixels, we choose to visualize the original outputs from all algorithms for fairness of comparison. The C-Shapley also yields the most interpretable results in CIFAR10. In particular, C-Shapley tends to mask the parts of head and body that distinguish deers and horses, and the human riding the horse. More visualization results are available in Appendix F.

### 5.3 EVALUATION WITH HUMAN SUBJECTS

We use human annotators on Amazon Mechanical Turk (AMT) to compare L-Shapley, C-Shapley and KernelSHAP on IMDB movie reviews. We aim to address two problems: (i) Are humans able to make a decision with top words alone? (ii) Are humans unable to make a decision with top words masked?

We randomly sample 200 movie reviews that are correctly classified by the model. Each review is assigned to five annotators. We ask humans on AMT to classify the sentiment of texts into five categories: strongly positive (+2), positive (+1), neutral (0), negative (-1), strongly negative (-2). See Appendix G for an example interface.

Texts have three types: (i) raw reviews; (ii) top ten words of each review ranked by L-Shapley, C-Shapley and KernelSHAP, where adjacent words, like "not satisfying or entertaining", keep their adjacency if selected simultaneously; and (iii) reviews with top words being masked. In the third type of texts, words are replaced with "[MASKED]" one after another, in the order produced by the respective algorithms, until the probability score of the correct class produced by the model is lower than 10%. We adopt the above design to make sure the majority of key words sensitive to the model have been masked. On average, around 14% of words in each review are masked for L-Shapley and C-Shapley, while 31.6% for KernelSHAP.

We measure the consistency (0 or 1) between the true labels and labels from human annotators, where a human label is positive if the average score over five annotators are larger than zero. Reviews with an average score of zero are neither put in the positive nor in the negative class. We also employ the standard deviation of scores on each review as a measure of disagreement between humans. Finally, the absolute value of the average scores from five annotators is used as a measure of confidence of decision.

The results of the two experiments are shown in Table 3. We observe humans become more consistent with the truth and more confident, and also have less disagreement with each other when they are presented with top words. Among the three algorithms, C-Shapley yields the highest performance in terms of consistency, agreement, and confidence. On the other hand, when top words are masked, humans are easier to make mistakes and are less certain about their judgement. L-Shapley harms the human judgement the most among the three algorithms, although KernelSHAP masks two times more words. The above experiments show that (i) Key words to the model convey an attitude toward a movie;, and (ii) Our algorithms find the key words more accurately.

## 6 DISCUSSION

We have proposed two new algorithms—L-Shapley and C-Shapley—for instancewise feature importance scoring, making use of a graphical representation of the data. We have demonstrated the superior performance of these algorithms compared to other methods on black-box models for instancewise feature importance scoring in both text and image classification with both quantitative metrics and human evaluation.

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

# A DETAILS OF DATA SETS AND MODEL STRUCTURE

## A.1 DATA SETS

**IMDB Review with Word-CNN**   The Internet Movie Review Dataset (IMDB) is a dataset of movie reviews for sentiment classification (Maas et al., 2011), which contains $50,000$ binary labeled movie reviews, with a split of $25,000$ for training and $25,000$ for testing.

**AG news with Char-CNN**   The AG news corpus is composed of titles and descriptions of $196,000$ news articles from $2,000$ news sources (Zhang et al., 2015). It is segmented into four classes, each containing $30,000$ training samples and $1,900$ testing samples.

**Yahoo! Answers with LSTM**   The corpus of Yahoo! Answers Topic Classification Dataset is divided into ten categories, each class containing $140,000$ training samples and $5,000$ testing samples. Each input text includes the question title, content and best answer.

**MNIST**   The MNIST data set contains $28 \times 28$ images of handwritten digits with ten categories $0 - 9$ (LeCun et al., 1998). A subset of MNIST data set composed of digits $3$ and $8$ is used for better visualization, with $12,000$ images for training and $1,000$ images for testing.

**CIFAR10**   The CIFAR10 data set (Krizhevsky, 2009) contains $32 \times 32$ images in ten classes. A subset of CIFAR10 data set composed of deers and horses is used for better visualization, with $10,000$ images for training and $2,000$ images for testing.

## A.2 MODEL STRUCTURE

**IMDB Review with Word-CNN**   The word-based CNN model is composed of a 50-dimensional word embedding, a 1-D convolutional layer of 250 filters and kernel size three, a max-pooling and a 250-dimensional dense layer as hidden layers. Both the convolutional and the dense layers are followed by ReLU as nonlinearity, and Dropout Srivastava et al. (2014) as regularization. The model is trained with rmsprop Hinton et al.. The model achieves an accuracy of $90.1\%$ on the test data set.

**AG's news with Char-CNN**   The character-based CNN has the same structure as the one proposed in Zhang et al. (2015), composed of six convolutional layers, three max-pooling layers, and two dense layers. The model is trained with SGD with momentum $0.9$ and decreasing step size initialized at $0.01$. (Details can be found in Zhang et al. (2015).) The model reaches accuracy of $90.09\%$ on the test data set.

**Yahoo! Answers with LSTM**   The network consists of a 300-dimensional randomly-initialized word embedding, a bidirectional LSTM, each LSTM unit of dimension 256, and a dropout layer as hidden layers. The model is trained with rmsprop Hinton et al.. The model reaches accuracy of $70.84\%$ on the test data set, close to the state-of-the-art accuracy of $71.2\%$ obtained by character-based CNN Zhang et al. (2015).

**MNIST**   A simple CNN model is trained on the data set, which achieves $99.7\%$ accuracy on the test data set. It is composed of two convolutional layers of kernel size $5 \times 5$ and a dense linear layer at last. The two convolutional layers contain 8 and 16 filters respectively, and both are followed by a max-pooling layer of pool size two.

**CIFAR10**   A convolutional neural network modified from AlexNet Krizhevsky et al. (2012) is trained on the subset. It is composed of six convolutional layers of kernel size $3 \times 3$ and two dense linear layers of dimension 512 and 256 at last. The six convolutional layers contain 48,48,96,96,192,192 filters respectively, and every two convolutional layers are followed by a max-pooling layer of pool size two and a dropout layer. The CNN model is trained with the Adam optimizer Kingma & Ba (2015) and achieves $96.1\%$ accuracy on the test data set.

# B  PROOF OF THEOREMS

In this appendix, we collect the proofs of Theorems 1 and 2.

## B.1  PROOF OF THEOREM 1

We state an elementary combinatorial equality required for the proof of the main theorem:

**Lemma 1** (A combinatorial equality). *For any positive integer $n$, and any pair of non-negative integers with $s \geq t$, we have*

$$\sum_{j=0}^{n} \frac{1}{\binom{n+s}{j+t}} \binom{n}{j} = \frac{s+1+n}{(s+1)\binom{s}{t}} \tag{13}$$

*Proof.* By the binomial theorem for negative integer exponents, we have

$$\frac{1}{(1-x)^{t+1}} = \sum_{j=0}^{\infty} \binom{j+t}{j} x^j.$$

The identity can be found by examination of the coefficient of $x^n$ in the expansion of

$$\frac{1}{(1-x)^{t+1}} \cdot \frac{1}{(1-x)^{s-t+1}} = \frac{1}{(1-x)^{s+1+1}}. \tag{14}$$

In fact, equating the coefficients of $x^n$ in the left and the right hand sides, we get

$$\sum_{j=0}^{n} \binom{j+t}{j}\binom{(n-j)+(s-t)}{n-j} = \binom{n+s+1}{n} = \frac{n+s+1}{s+1}\binom{n+s}{n}. \tag{15}$$

Moving $\binom{n+s}{n}$ to the right hand side and expanding the binomial coefficients, we have

$$\sum_{j=0}^{n} \frac{(j+t)!}{j!t!} \cdot \frac{(n-j+s-t)!}{(n-j)!(s-t)!} \cdot \frac{n!s!}{(n+s)!} = \frac{n+s+1}{s+1}, \tag{16}$$

which implies

$$\sum_{j=0}^{n} \binom{n}{j}\binom{s}{t} \bigg/ \binom{n+s}{j+t} = \sum_{j=0}^{n} \frac{n!}{(n-j)!j!} \cdot \frac{s!}{t!(s-t)!} \cdot \frac{((n+s)-(j+t))!(j+t)!}{(n+s)!}$$

$$= \sum_{j=0}^{n} \frac{(j+t)!}{j!t!} \cdot \frac{(n-j+s-t)!}{(n-j)!(s-t)!} \cdot \frac{n!s!}{(n+s)!} = \frac{n+s+1}{s+1}.$$

$\square$

Taking this lemma, we now prove the theorem. We split our analysis into two cases, namely $S = \mathcal{N}_k(i)$ versus $S \subset \mathcal{N}_k(i)$. For notational convenience, we extend the definition of L-Shapley estimate for feature $i$ to an arbitrary feature subset $S$ containing $i$. In particular, we define

$$\hat{\phi}_x^S(i) := \frac{1}{|S|} \sum_{\substack{T \ni i \\ T \subseteq S}} \frac{1}{\binom{|S|-1}{|T|-1}} m_x(T, i). \tag{17}$$

**Case 1:**  First, suppose that $S = \mathcal{N}_k(i)$. For any subset $A \subset [d]$, we introduce the shorthand notation $U_S(A) := A \cap S$ and $V_S(A) := A \cap S^c$, and note that $A = U_S(A) \cup V_S(A)$. Recalling the definition of the Shapley value, let us partition all the subsets $A$ based on $U_S(A)$, in particular writing

$$\phi_X(i) = \frac{1}{d} \sum_{\substack{A \subseteq [d] \\ A \ni i}} \frac{1}{\binom{d-1}{|A|-1}} m_X(A, i) = \frac{1}{d} \sum_{\substack{U \subseteq S \\ U \ni i}} \sum_{\substack{A \subseteq [d] \\ U_S(A)=U}} \frac{1}{\binom{d-1}{|A|-1}} m_X(A, i).$$

Based on this partitioning, the expected error between $\hat{\phi}_X^S(i)$ and $\phi_X(i)$ can be written as

$$\mathbb{E}\left|\hat{\phi}_X^S(i) - \phi_X(i)\right| = \mathbb{E}\left|\frac{1}{|S|}\sum_{\substack{U \subseteq S \\ U \ni i}}\frac{1}{\binom{|S|-1}{|U|-1}}m_X(U,i) - \frac{1}{d}\sum_{\substack{U \subseteq S \\ U \ni i}}\sum_{\substack{A \subseteq [d] \\ U_S(A)=U}}\frac{1}{\binom{d-1}{|A|-1}}m_X(A,i)\right|. \quad (18)$$

Partitioning the set $\{A : U_S(A) = U\}$ by the size of $V_S(A) = A \cap S^c$, we observe that

$$\sum_{\substack{A \subseteq [d] \\ U_S(A)=U}}\frac{1}{\binom{d-1}{|A|-1}} = \sum_{i=0}^{d-|S|}\frac{1}{\binom{d-1}{i+|U|-1}}\binom{d-|S|}{i}$$

$$= \frac{(|S|-1)+1+(d-|S|)}{((|S|-1)+1)\binom{|S|-1}{|U|-1}}$$

$$= \frac{d}{|S|}\frac{1}{\binom{|S|-1}{|U|-1}},$$

where we have applied Lemma 1 with $n = d - |S|$, $s = |S| - 1$, and $t = |U| - 1$. Substituting this equivalence into equation equation 18, we find that the expected error can be upper bounded by

$$\mathbb{E}|\hat{\phi}_X^S(i) - \phi_X(i)| \leq \frac{1}{d}\sum_{\substack{U \subseteq S \\ U \ni i}}\sum_{\substack{A \subseteq [d] \\ U_S(A)=U}}\frac{1}{\binom{d-1}{|A|-1}}\mathbb{E}\left|m_X(U,i) - m_X(A,i)\right|, \quad (19)$$

where we recall that $A = U_S(A) \cup V_S(A)$.

Now omitting the dependence of $U_S(A), V_S(A)$ on $A$ for notational simplicity, we now write the difference as

$$m_X(A,i) - m_X(U,i) = \mathbb{E}_m\left[\log\frac{\mathbb{P}_m(Y|X_{U \cup V})}{\mathbb{P}_m(Y|X_{U \cup V \setminus \{i\}})} - \log\frac{\mathbb{P}_m(Y|X_U)}{\mathbb{P}_m(Y|X_{U \setminus \{i\}})} \mid X\right]$$

$$= \mathbb{E}_m\left[\log\frac{\mathbb{P}(Y,X_{U \setminus \{i\}})\mathbb{P}(X_U)P(X_{U \cup V \setminus \{i\}})P(X_{U \cup V},Y)}{\mathbb{P}(Y,X_U)\mathbb{P}(X_{U \setminus \{i\}})P(X_{U \cup V})P(X_{U \cup V \setminus \{i\}},Y)} \mid X\right]$$

$$= \mathbb{E}_m\left[\log\frac{\mathbb{P}(X_i,X_V \mid X_{U \setminus \{i\}},Y)}{\mathbb{P}(X_i \mid X_{U \setminus \{i\}},Y)\mathbb{P}(X_V \mid X_{U \setminus \{i\}},Y)}\right.$$

$$\left. - \log\frac{\mathbb{P}(X_i,X_V|X_{U \setminus \{i\}})}{\mathbb{P}(X_i|X_{U \setminus \{i\}})\mathbb{P}(X_V \mid X_{U \setminus \{i\}})} \mid X\right].$$

Substituting this equivalence into our earlier bound equation 19 and taking an expectation over $X$ on both sides, we find that the expected error is upper bounded as

$$\mathbb{E}\left|\hat{\phi}_X^S(i) - \phi_X(i)\right| \leq \frac{1}{d}\sum_{\substack{U \subseteq S \\ U \ni i}}\sum_{\substack{A \subseteq [d] \\ U_S(A)=U}}\frac{1}{\binom{d-1}{|A|-1}}\left\{\mathbb{E}\left|\log\frac{\mathbb{P}(X_i,X_{V_S(A)}|X_{U \setminus \{i\}},Y)}{\mathbb{P}(X_i|X_{U \setminus \{i\}},Y)\mathbb{P}(X_{V_S(A)}|X_{U \setminus \{i\}},Y)}\right|\right.$$

$$\left. + \mathbb{E}\left|\log\frac{\mathbb{P}(X_i,X_{V_S(A)}|X_{U \setminus \{i\}})}{\mathbb{P}(X_i \mid X_{U \setminus \{i\}})\mathbb{P}(X_{V_S(A)}|X_{U \setminus \{i\}})}\right|\right\}.$$

Recalling the definition of the absolute mutual information, we see that

$$\mathbb{E}|\hat{\phi}_X^S(i) - \phi_X(i)| \leq \frac{1}{d}\sum_{\substack{U \subseteq S \\ U \ni i}}\sum_{\substack{A \subseteq [d] \\ U_S(A)=U}}\frac{1}{\binom{d-1}{|A|-1}}\left\{I_a(X_i; X_{V_S(A)} \mid X_{U \setminus \{i\}},Y)\right.$$

$$\left. + I_a(X_i; X_{V_S(A)} \mid X_{U \setminus \{i\}})\right\}$$

$$\leq 2\varepsilon,$$

which completes the proof of the claimed bound.

Finally, in the special case that $X_i \perp\!\!\!\perp X_{[d]\setminus S}|X_T$ and $X_i \perp\!\!\!\perp X_{[d]\setminus S}|X_T, Y$ for any $T \subset S$, then this inequality holds with $\varepsilon = 0$, which implies $\mathbb{E}|\hat{\phi}_X^S(i) - \phi_X(i)| = 0$. Therefore, we have $\hat{\phi}_X^S(i) = \phi_X(i)$ almost surely, as claimed.

**Case 2:**  We now consider the general case in which $S \subset \mathcal{N}_k(i)$. Using the previous arguments, we can show

$$\mathbb{E}|\hat{\phi}_X^S(i) - \phi_X^k(i)| \leq 2\varepsilon, \quad \text{and} \quad \mathbb{E}|\hat{\phi}_X^S(i) - \phi_X(i)| \leq 2\varepsilon.$$

Appylying the triangle inequality yields $\mathbb{E}|\hat{\phi}_X^k(i) - \phi_X(i)| \leq 4\varepsilon$, which establishes the claim.

### B.2  PROOF OF THEOREM 2

As in the previous proof, we divide our analysis into two cases.

**Case 1:**  First, suppose that $S = \mathcal{N}_k(i) = [d]$. For any subset $A \subset S$ with $i \in A$, we can partition $A$ into two components $U_S(A)$ and $V_S(A)$, such that $i \in U_S(A)$ and $U_S(A)$ is a connected subsequence. $V_S(A)$ is disconnected from $U_S(A)$. We also define

$$\mathcal{C} = \{U \mid i \in U, U \subset [d], U \text{ is a connected subsequence.}\} \tag{20}$$

We partition all the subsets $A \subset S$ based on $U_S(A)$ in the definition of the Shapley value:

$$\phi_X(i) = \frac{1}{d} \sum_{\substack{A \subseteq S \\ A \ni i}} \frac{1}{\binom{d-1}{|A|-1}} m_X(A, i)$$

$$= \frac{1}{d} \sum_{U \in \mathcal{C}} \sum_{A : U_S(A) = U} \frac{1}{\binom{d-1}{|A|-1}} m_X(A, i).$$

The expected error between $\tilde{\phi}_X^{[d]}(i)$ and $\phi_X(i)$ is

$$\mathbb{E}|\tilde{\phi}_X^{[d]}(i) - \phi_X(i)| = \mathbb{E}\left| \frac{1}{d} \sum_{U \in \mathcal{C}} \frac{2d}{(|U|+2)(|U|+1)|U|} m_X(U, i) - \frac{1}{d} \sum_{U \in \mathcal{C}} \sum_{A : U_S(A) = U} \frac{1}{\binom{d-1}{|A|-1}} m_X(A, i) \right|. \tag{21}$$

Partitioning $\{A : U_S(A) = U\}$ by the size of $V_S(A)$, we observe that

$$\sum_{A : U_S(A) = U} \frac{1}{\binom{d-1}{|A|-1}} = \sum_{i=0}^{d-|U|-2} \frac{1}{\binom{d-1}{i+|U|-1}} \binom{d-|U|-2}{i}$$

$$= \frac{(|U|+1) + 1 + (d-|U|-2)}{((|U|+1)+1)\binom{|U|+1}{|U|-1}}$$

$$= \frac{2d}{(|U|+2)(|U|+1)|U|}, \tag{22}$$

where we apply Lemma 1 with $n = d - |U| - 2$, $s = |U| + 1$ and $t = |U| - 1$. From equation equation 21, the expected error can be upper bounded by

$$\mathbb{E}\left| \tilde{\phi}_X^{[d]}(i) - \phi_X(i) \right| \leq \frac{1}{d} \sum_{U \in \mathcal{C}} \sum_{A : U_S(A) = U} \frac{1}{\binom{d-1}{|A|-1}} \mathbb{E}\left| m_X(U, i) - m_X(A, i) \right|,$$

where $A = U_S(A) \cup V_S(A)$. We omit the dependence of $U_S(A)$ and $V_S(A)$ on the pair $(A, S)$ for notational simplicity, and observe that the difference between $m_x(A, i)$ and $m_x(U, i)$ is

$$
\begin{aligned}
m_X(A, i) - m_X(U, i) &= \mathbb{E}_m \left[ \log \frac{\mathbb{P}_m(Y|X_{U \cup V})}{\mathbb{P}_m(Y|X_{U \cup V \setminus \{i\}})} - \log \frac{\mathbb{P}_m(Y|X_U)}{\mathbb{P}_m(Y|X_{U \setminus \{i\}})} \mid X \right] \\
&= \mathbb{E}_m \left[ \log \frac{\mathbb{P}(Y, X_{U \setminus \{i\}}) \mathbb{P}(X_U) P(X_{U \cup V \setminus \{i\}}) P(X_{U \cup V}, Y)}{\mathbb{P}(Y, X_U) \mathbb{P}(X_{U \setminus \{i\}}) P(X_{U \cup V}) P(X_{U \cup V \setminus \{i\}}, Y)} \mid X \right] \\
&= \mathbb{E}_m \left[ \log \frac{\mathbb{P}(X_i, X_V | X_{U \setminus \{i\}}, Y)}{\mathbb{P}(X_i | X_{U \setminus \{i\}}, Y) \mathbb{P}(X_V | X_{U \setminus \{i\}}, Y)} \right. \\
&\qquad \left. - \log \frac{\mathbb{P}(X_i, X_V | X_{U \setminus \{i\}})}{\mathbb{P}(X_i | X_{U \setminus \{i\}}) \mathbb{P}(X_V | X_{U \setminus \{i\}})} \mid X \right].
\end{aligned}
$$

Taking an expectation over $X$ at both sides, we can upper bound the expected error by

$$
\begin{aligned}
\mathbb{E}|\tilde{\phi}_X^{[d]}(i) - \phi_X(i)| &\leq \frac{1}{d} \sum_{U \in \mathcal{C}} \sum_{A: U_S(A) = U} \frac{1}{\binom{d-1}{|A|-1}} \left( \mathbb{E} \left| \log \frac{\mathbb{P}(X_i, X_{V_S(A)} | X_{U \setminus \{i\}}, Y)}{\mathbb{P}(X_i | X_{U \setminus \{i\}}, Y) \mathbb{P}(X_{V_S(A)} | X_{U \setminus \{i\}}, Y)} \right| \right. \\
&\qquad \left. + \mathbb{E} \left| \log \frac{\mathbb{P}(X_i, X_{V_S(A)} | X_{U \setminus \{i\}})}{\mathbb{P}(X_i | X_{U \setminus \{i\}}) \mathbb{P}(X_{V_S(A)} | X_{U \setminus \{i\}})} \right| \right) \\
&= \frac{1}{d} \sum_{U \in \mathcal{C}} \sum_{A: U_S(A) = U} \frac{1}{\binom{d-1}{|A|-1}} (I_a(X_i; X_{V_S(A)} | X_{U \setminus \{i\}}, Y) + I_a(X_i; X_{V_S(A)} | X_{U \setminus \{i\}})) \\
&\leq 2\varepsilon.
\end{aligned}
$$

Let $R(U) := [d] - U \cup \{\max(u - 1, 1), \min(u + l + 1, d)\}$. If we have $X_i \perp\!\!\!\perp X_{R(U)} | X_{U \setminus \{i\}}$ and $X_i \perp\!\!\!\perp X_{R(U)} | X_{U \setminus \{i\}}, Y$ for any $U \subset [d]$, then $\varepsilon = 0$, which implies $\mathbb{E}|\tilde{\phi}_X^{[d]}(i) - \phi_X(i)| = 0$. Therefore, we have $\tilde{\phi}_X^{[d]}(i) = \phi_X(i)$ almost surely.

**Case 2:** We now turn to the general case $S \subset \mathcal{N}_k(i) \subset [d]$. Similar as above, we can show

$$
\mathbb{E}|\tilde{\phi}_X^k(i) - \hat{\phi}_X^k(i)| \leq 2\varepsilon.
$$

Based on Theorem 1, we have

$$
\mathbb{E}|\hat{\phi}_X^k(i) - \phi_X(i)| \leq 4\varepsilon.
$$

Applying the triangle yields $\mathbb{E}|\tilde{\phi}_X^k(i) - \phi_X(i)| \leq 6\varepsilon$, which establishes the claim.

## C    RANK CORRELATION WITH THE SHAPLEY VALUE

We address the problem of how the rank of features produced by various approximation algorithms correlates with the rank produced by the true Shapley value. We sample a subset of test data from Yahoo! Answers with 9-12 words, so that the underlying Shapley scores can be accurately computed. We employ two common metrics, Kendall's $\tau$ and Spearman's $\rho$ (Kendall, 1975), to measure the similarity (correlation) between two ranks.

The result is shown in Figure 5. The rank correlation between L-Shapley and the Shapley value is the highest, followed by C-Shapley, consistent across both of the two metrics. Given the limited length of each instance, the search space for sampling based algorithms is relatively small. Thus there is only a slight performance gain of our algorithms over KernelSHAP and SampleShapley.

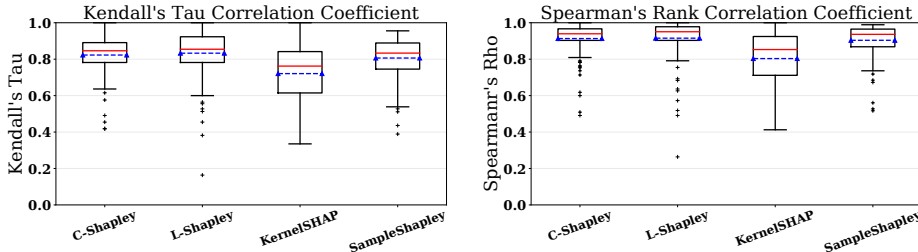

Figure 5: The box plots of Kendall's $\tau$ and Spearman's $\rho$ between various algorithms (with the same computational complexity) and the Shapley value. The red line and the dotted blue line on each box are the median and the mean respectively. (Higher is better.)

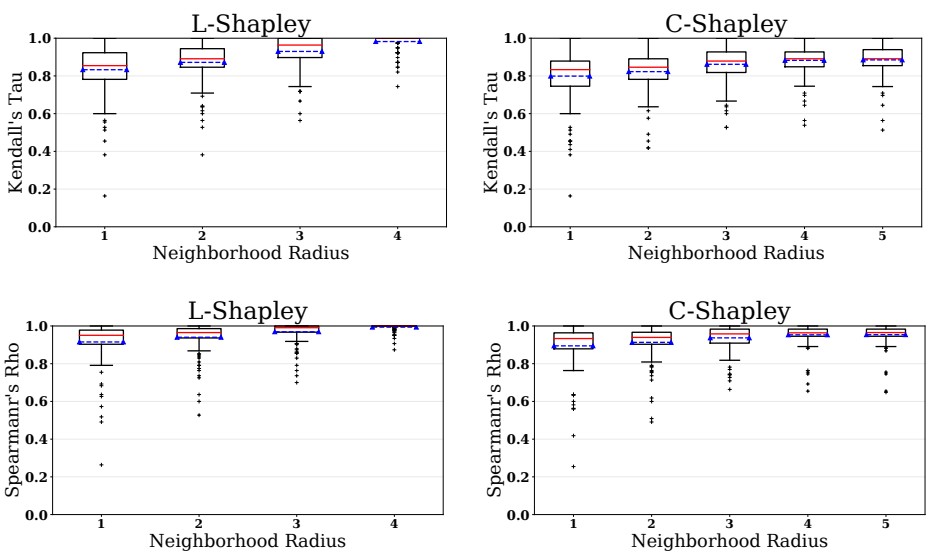

Figure 6: Kendall's $\tau$ and Spearman's $\rho$ between L-Shapley and the Shapley value, C-Shapley and the Shapley value vs. the neighborhood radius. The red line and the dotted blue line on each box are the median and the mean respectively. (Higher is better.)

## D    SENSITIVITY OF L-SHAPLEY AND C-SHAPLEY

We study the sensitivity of L-Shapley and C-Shapley to the radius of neighborhood on the subsampled data from Yahoo! Answers in Appendix C. We employ Kendall's $\tau$ and Spearman's $\rho$ (Kendall,

1975) to measure the rank correlation between scores from the proposed methods and the Shapley value[2].

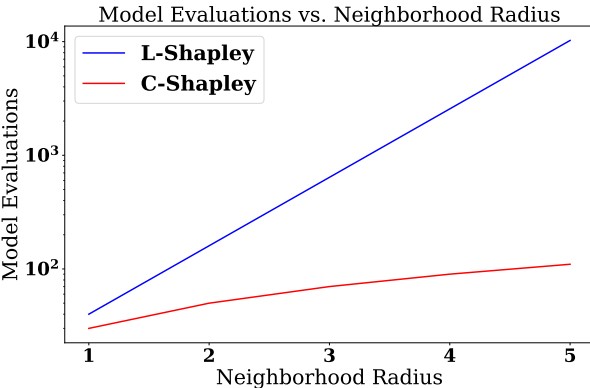

Figure 7: Number of model evaluations vs. neighborhood radius (in log scale) on an instance with ten features.

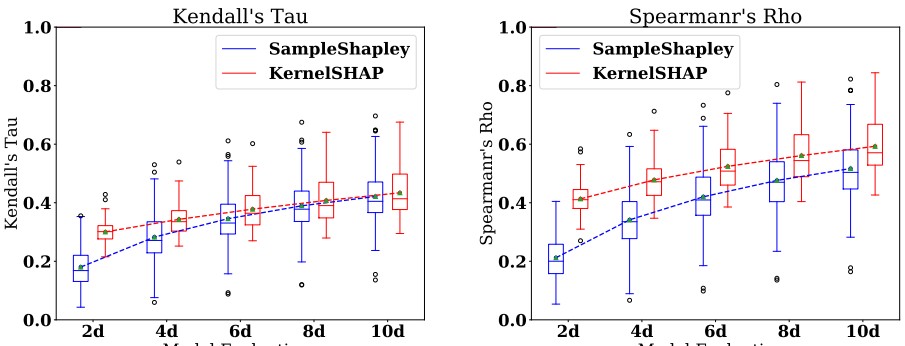

Figure 8: The box plots of average pairwise Kendall's $\tau$ and Spearman's $\rho$ vs. the number of model evaluations over 30 replicates. (Higher is better.) Dash lines (with green dots) plots the mean, and the solid lines in box plots are medians. The number of features $d$ varies across different instances.

Figure 6 shows how Kendall's $\tau$ and Spearman's $\rho$ between the proposed algorithms and the Shapley value vary with the radius of neighborhood. We observe the bias of the proposed algorithms decreases gradually with increasing neighborhood radius. Figure 7 plots the number of model evaluations as a function of neighborhood radius for both algorithms, on an example instance with ten features[3]. The complexity of L-Shapley grows exponentially with the neighborhood radius while the complexity of C-Shapley grows linearly.

# E  VARIABILITY OF SAMPLING BASED ALGORITHMS

We empirically evaluate the variance of SampleShapley and KernelSHAP in the setting where the sample size is linear in the number of features. The experiment is carried out on the test data set of IMDB movie reviews. For each method, we run 30 replications on every instance, which generates 30 scores. Given the varied scalability of underlying Shapley values, we again seek a nonparametric approach to measure the variability of sampling based algorithms. On each instance, we compute

---

[2]The nonparametric rank correlation is used instead of Pearson correlation coefficient because of the violation of identical assumption across different instances

[3]Model evaluations can be easily paralleled on a modern GPU. Hence we plot the number of model evaluations instead of real running time, which depends on the availability of computational resource.

the pairwise Kendall's $\tau^4$ and Spearman's $\rho$ among the 30 runs of a single method. Then we use the average of $\binom{30}{2}$ $\tau$s and $\rho$s respectively as measures of statistical dispersion[5].

Figure 8 shows the variability of SampleShapley and KernelSHAP as a function of the number of model evaluations. The ticks $2d, 4d, \ldots$ on the x-axis are the number of model evaluations, where $d$ is the number of features which varies across different instances. As a concrete example, on the rightmost box plot, KernelSHAP carries out $10d = 2,000$ model evaluations on an instance with $d = 200$ features.

# F    VISUALIZATION OF MNIST AND CIFAR10

## F.1    MNIST

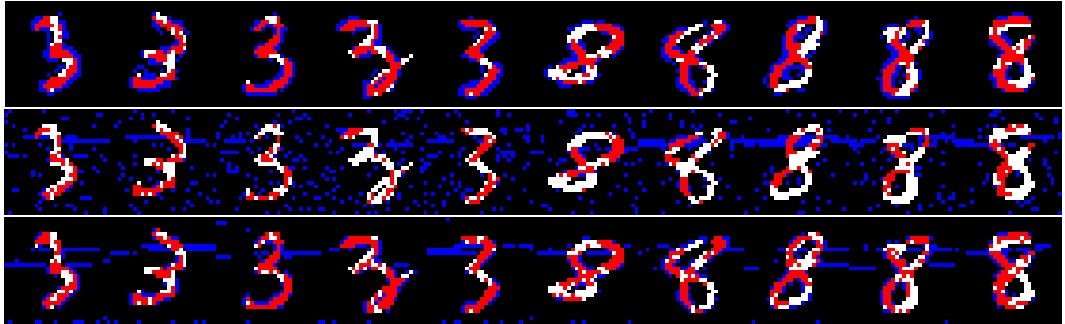

Figure 9: The above figure shows explanation results on ten randomly selected figures of 3 and 8. We mask 118 pixels out of 784 pixels for each image, where the masked pixels are colored with red if activated (white) and blue otherwise. The masking scores are produced by ConnectedShapley, KernelSHAP and SampleShapley for each row respectively.

## F.2    CIFAR10

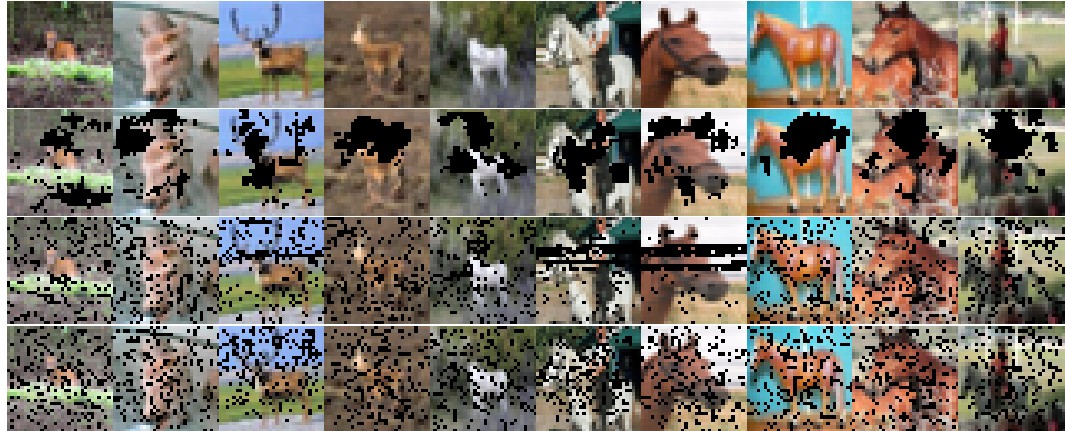

Figure 10: The above figure shows explanation results on ten randomly selected figures of horses and deers. We mask 205 top pixels chosen by each method out of 1,024 pixels for each image. The first row shows the original images. The rest of the rows show images masked based on scores produced by ConnectedShapley, KernelSHAP and SampleShapley respectively.

---

[4]$\tau$-b version is used which can account for ties (Kendall, 1975).

[5]The former can be linked to the variance when one models permutation with the Mallows Model. A discussion can be found in (Jiao & Vert, 2018)

# G  HUMEN EVALUATION INTERFACE

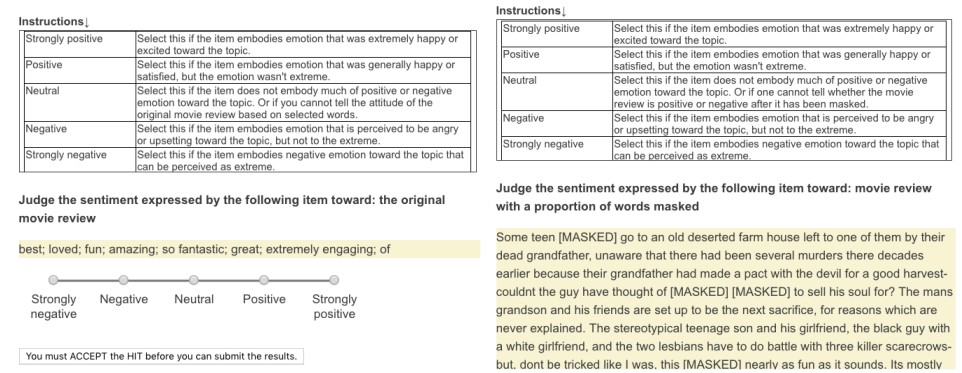

Figure 11: Interfaces of Amazon Mechanical Turk where annotators are asked to infer the sentiment of the original reviews based on selected words and masked reviews.

# H VISUALIZATION ON IMDB WITH WORD-CNN

Only the ten words with the largest scores and the ten words with the smallest scores are colorized. The words with largest scores with respect to the predicted class are highlighted with red. The ten words with smallest scores with respect to the predicted class are highlighted with blue. (In other words, red words tend to contain positive attitude for a positive prediction, but negative attitude for a negative prediction.) The corresponding RGB entries are linearly interpolated based on importance scores. The lighter a color is, the less information with respect to the prediction the corresponding word is.

Table 4: Visualization on IMDB with Word-CNN

| Class | Perturbed |
| --- | --- |
| positive | This was the second Cinemascope spectacle that Fox produced after the Robe . Notice how some of the Roman sets are redressed to pass for Egyptian sets . The film is produced with all first class elements , beautiful photography , stirring soundtrack ( Alfred Newman and Bernard Herrmann _ see if you can tell which composer scored specific scenes ). However , the principal acting is a bit weak . Edmund Purdom seems to have a limited range of emotions and is uninteresting to watch . The best performances come from Peter Ustinov as the one _ eyed slave and Polish actress Bella Darvi as the Babylonian temptress Nefer . I find this movie in general to be strong on plot which is rare for these large spectacles produced at the time . All in all , the film does an interesting and entertaining job of social commentary on what Egyptian society might have looked like . |
| negative | I saw this movie only because Sophie Marceau . However , her acting abilities its no enough to salve this movie . Almost all cast dont play their character well , exception for Sophie and Frederic . The plot could give a rise a better movie if the right pieces was in the right places . I saw several good french movies but this one i dont like . |
| negative | If it wasnt for the performances of Barry Diamond and Art Evans as the clueless stoners , I would have no reason to recommend this to anyone . The plot centers around a 10 year high school reunion , which takes place in a supposed abandon high school ( looks more like a prop from a 1950s low budget horror flick ), and the deranged student the class pulled a very traumatizing prank on . This student desires to kill off the entire class for revenge . John Hughes falls in love with his characters too much , as only one student is killed as well as the lunch lady ( Goonies Anne Ramsey ). Were led to believe that the horny coupled gets killed , but never see a blasted thing ! This is a horrible movie that continued National Lampoons downward spiral throughout the 80s and 90s . |

| positive | This is a wonderful look , you should pardon the pun , at 22 women talking about breasts __ theirs , their mothers , other womens , and how they affect so many aspects of their lives . Young girls , old women , and everyone in between ( with all shapes , sizes , configurations , etc ) talk about developing , reacting , celebrating , hiding , enhancing , or reducing their breasts . Its charming , delightful , sad , funny , and everything in between . Intercut with documentary footage and clips from those famous old young womens films that the girls got taken to the cafeteria to see , the interviews are a fascinating window for men who love women & their breasts into what the other half has to say when they dont know youre listening . |
| --- | --- |
| positive | This movie doesnt have any pretense at being great art , which is good . But it is a well written script with well developed characters and solid acting . I think if I wrote it I could do without the drama surrounding the wife , but it wasnt distracting enough to detract from the main story concerning Minnie Drivers character . I think that all too often Hollywood abandons an attempt at real quality writing to try and inject more visual drama when , with an adult themed movie such as this , the emotional type of drama is all thats really needed _ and probably more believable too . Overall , its a very well done offering and well worth seeing . |
| positive | This is just as good as the original 101 if not better . Of course , Cruella steals the show with her outrageous behaviour and outfits , and the movie was probably made because the public wanted to see more of Cruella . We see a lot more of her this time round . I also like Ioan Gruffudd as Kevin , the rather bumbling male lead . To use Paris as the climax of the movie was a clever idea . The movie is well worth watching whatever your age , provided you like animals . |
| negative | I vowed a long time ago to NEVER , EVER watch a movie that has ANYONE who EVER was a regular cast member of Saturday Night Live . I didnt rent Corky Romano but I was forced by my unfailing good manners to watch it for half an hour . Then my good manners failed . Stupid , not funny . Tedious , not hilarious . Bad , not good . That in a nutshell is all I can say for this video . |
| positive | this took me back to my childhood in the 1950 s so corny but just fab no one ever could play FLASH GORDON like LARRY BUSTER CRABBE , just great . i have two more series to view flash gordons trip to mars and flash gordon conquers the universe cannot wait |
| positive | Magnificent and unforgettable , stunningly atmospheric , and brilliantly acted by all . I really cannot understand what sort of people are panning this masterpiece and giving the preponderance of votes as 8 ( and nine ones !) This , along with Grapes of Wrath , is John Fords greatest movie . I would say that Long Voyage Home is next in line , though quite a way back . Rating : 10 . It deserves a 12 . |

| | |
|---|---|
| negative | Yes , In 35 years of film going I have finally viewed the stinker that surpasses all other ghastly movies I have seen . Beating Good Will Hunting Baise Moi and Flirt for sheer awfulness . This is pretentious blige of the first order ... not even entertaining pretentious bilge . The effects are cheap , and worse _ pointless . The script seems to have been written by a first year film student who doesnt get out much but wants to appear full of portent ! The acting is simply undescribably bad _ Tilda Swinton caps a career filled with vacuous woodeness with a performance which veers neurotically between comotose and laughable intensity . Apparently , some fool out there has allowed the director of this film to make another one ... be warned |
| positive | I find it sad that just because Edward Norton did not want to be in the film or have anything to do with it , people automatically think the movie sucks without even watching it or giving it a chance . I really hope Norton did not do this . He is a fine actor and all but he scared people away from a decent movie . I found it entertaining . It wasnt mind blowing or anything with crazy special effects , but it was not a bad . It was fun to watch . But yea , definitely not a bad / horrible movie . 7 / 10 |
| positive | Beautifully done . A lot of angst . Friendship may not endure all , but in the end its all that matters , or so a group of friends learn . I have watched it over and over again . The music is also amazing . When Kei loses the one friend he has he gives up until he meets Sho , an orphan boy who is not repelled by his true nature . In the lawless streets of Mallepa they struggle for their own place among a melting pot of Asian races , and learn that sometimes being on top can cost you more than you are ever ready to pay . A surprise ending that grips as much as the whole movie does . I couldnt get enough of it . Gackt and Hyde do a wonderful job of acting , proving they are more than pretty boys who sing . |
| positive | Ive watched the first 17 episodes and this series is simply amazing ! I havent been this interested in an anime series since Neon Genesis Evangelion . This series is actually based off an h _ game , which Im not sure if its been done before or not , I havent played the game , but from what Ive heard it follows it very well . I give this series a 10 / 10 . It has a great story , interesting characters , and some of the best animation Ive seen . It also has some great Japanese music in it too ! If you havent seen this series yet , check it out . You can find subbed episodes on some anime websites out there , its straight out of Japan . |

| negative | Based on a Ray Bradbury story ; a professional photographer ( Brian Kerwin ) returns to his modest home near a tiny desert town , where most of the citizens wishes he stayed away . A lonely boy ( Jonathan Carrasco ) latches onto him for the attention ; and the two witness the landing of an alien craft in the rocky region of the desert . The aliens turn themselves into the images of townspeople . Kerwin must convince evacuation of the town and falls in love with the young boys mother ( Elizabeth Pena ). Acting is pretty shallow ; the story line is no worse than some others ; this movie leaves you feeling that you got shorted on a decent ending . Supporting cast includes : Howard Morris , Dean Norris and Mickey Jones . |
|---|---|
| negative | very badly made film , the action / violence scenes are ridiculous . 1 point for the presence of Burton and Mastroianni + 1 point for the real tragic event of the massacre of the innocent italians : 2 / 10 . |
| negative | J Carol Nash and Ralph Morgan star in a movie about a mad scientist in love with a pianists daughter . When his advances are spurned he injects the father with a disfiguring disease so that she will be forced to come to him to get a cure . God this is awful . Its dull and boring and youll nod off before the pianist gets uglified , I was on the verge . Yea it picks up once things are set in motion but this is one of those old movies better remembered then seen again . If you must see it come in late4 out of 10 |
| positive | GEORGE LOPEZ , in my opinion , is an absolute ABC classic ! I havent seen every episode , but I still enjoy it . There are many episodes that I enjoyed . One of them was where Amy ( Sandra Bullock ) walked into a moving piece of machinery . If you want to know why , youll have to have seen it for yourself . Before I wrap this up , Id like to say that everyone always gave a good performance , the production design was spectacular , the costumes were well _ designed , and the writing was always very strong . In conclusion , even though new episodes can currently be seen , I strongly recommend you catch it just in case it goes off the air for good . |
| positive | Here is a movie of adventure , determination , heroism , & bravery . Plus , its set back in the late 1800s which makes it even more interesting . Its a wonderful , adventurous storyline , and Alyssa Milano is wonderful at playing the wholesome , confident , no _ nonsense Fizzy ... a great role _ model . This is one of my favorite movies . It is a movie to be watched again and again and will inspire you and enrich your life without a doubt . Not only is the storyline excellent , but the movie also has fabulous scenery and music and is wonderfully directed . This movie is as good as gold ! |

| | |
|---|---|
| negative | This crock of doodoo won a award ? They must have been desperate for giving out an award for something . This movie reeks of teeny bopper stuff and it made me sick . Thankfully I watched it alongside MST3Ks Mike and the bots so it made it bearable . Horrid acting , unsettling mother / daughter moment , silly premise , if you want a bad movie here it is . Be warned though watch it with Mike and the bots or you will suffer . 1 out of 10 . I still cant believe it won an award , and the director is defending this movie ! |
| negative | I had never heard of this one before it turned up on Cable TV . Its very typical of late 50s sci _ fi : sober , depressing and not a little paranoid ! Despite the equally typical inclusion of a romantic couple , the film is pretty much put across in a documentary style _ which is perhaps a cheap way of leaving a lot of the exposition to narration and an excuse to insert as much stock footage as is humanly possibly for what is unmistakably an extremely low _ budget venture ! While not uninteresting in itself ( the _ apocalypse _ via _ renegade _ missile angle later utilized , with far greater aplomb , for both DR . STRANGELOVE [ 1964 ] and FAIL _ SAFE [ 1964 ]) and mercifully short , the films single _ minded approach to its subject matter results in a good deal of unintentional laughter _ particularly in the scenes involving an imminent childbirth and a gang of clueless juvenile delinquents ! |
| positive | There have been several films about Zorro , some even made in Europe , e . g . Alain Delon . This role has also been played by outstanding actors , such as Tyrone Power and Anthony Hopkins , but to me the best of all times has always been Reed Hadley . This serial gives you the opportunity to see an interesting western , where you will only discover the real villain , Don del Oro , at its end . The serial also has good performance of various actors of movies B like Ed Cobb , ex _ Tarzan Jim Pierce , C . Montague Shaw , eternal villains like John Merton and Charles King , and a very good performance of Hadley as Zorro . He was quick , smart , used well his whip and sword , and his voice was the best for any Zorro . |
| negative | Well it certainly stunned me _ I can not believe that someone made another Australian film thats even more boring than Somersault . The story is implausible , the characters , with the exception of Friels and Mailmans characters , are unlikeable and wooden , Tom Long possesses a VAST array of facial expressions : happy and not happy , and the sex scenes , which could have been very confronting and disturbingly erotic , would have been at home in a low _ budget porno flick . This is the first movie I have seen in 30 years of cinema _ going that has had me on the edge of my seat .... ready to get up and leave . The best thing about this movie is the promotional poster . |

| positive | The main reason I loved this movie is because IMx ( formerly Immature ) were in it . They were in House Party 3 when they were 11 , but they are all grown up now ! I was a little shocked at some of the things they were doing in the movie ( almost ready to tear my hair out ), but I had to realize that they were not my little boys anymore . I think Chris Stokes did a pretty good job , considering that is was his first movie . |
|---|---|
| positive | Making this short and to the point . This movie was great ! I loved it ! I actually picked this up at a Hollywood Video for 3 bucks on VHS and watched it about 5 times in the last couple weeks . Im a big Bogart fan and I just latched onto this movie . I thought the song was funny and now have it as a ring tone on my phone . Robert Sacchi is great and pulls off a good Bogart . His nose is a little big , his voice is a Bogart _ Columbo mix , and he does a few things that are awkward but otherwise , he was fantastic and this film was wonderful . No one can be a perfect Bogart but he was great . Remember , Sam Marlow is a fan of Bogart and isnt going to do everything he did . He mentions a lot of other movies and does some things that were never part of the real Bogarts characters . But , its so funny and hilarious and has a great cast , including some beautiful women . Watch it and have fun ! |
| positive | Otto Premingers Dana Andrews cycle of films noirs are among the ( largely ) unsung jewels of the genre . Because they lack paranoia , misogyny or hysteria , they may have seemed out of place at the time , but the clear _ eyed imagery , the complex play with identity , masculinity and representation , the subversion of traditional psychological tenets , the austere , geometrical style all seem startlingly modern today , and very similar to Melville . The lucid ironies of this film are so loaded , brutal and ironic that the happy ending is one of the cruellest in Hollywood history . Brilliant on the level of entertaining thriller as well , tense , and packed with double _ edged dialogue . |

Table 4: Visualization on IMDB with Word-CNN.

# I   VISUALIZATION ON YAHOO! ANSWERS WITH LSTM

Only the ten words with the largest scores and the ten words with the smallest scores are colorized. The words with largest scores with respect to the predicted class are highlighted with red. The ten words with smallest scores with respect to the predicted class are highlighted with blue. The corresponding RGB entries are linearly interpolated based on importance scores. The lighter a color is, the less information with respect to the prediction the corresponding word is.

Table 5: Visualization on Yahoo! Answers with LSTM

| Class | Perturbed |
|---|---|
| Society, Culture | eve was the mother of cain and abel did she have any daughters yes read genesis 5 |
| Family, Relationships | good guys why is it that women leave me because they say i am to nice please tell me they want some one who is disrespectful and who will use them there dad was probably like that so that is what they are use to its normal to them |
| Science, Mathematics | what effect may global warming have on britain it might rain less longer summers not so bloody freezing in winter oh and the small matter of maybe wales flooding n n n nso this global warming is a bad thing yeah |
| Politics, Government | so if our borders need fixing and let's agree that they do how do we pay for it the united states congress seems to come up with all kinds of money for a lot of silly things here are some examples n 75 000 for seafood waste research n 500 000 for the arctic winter games n 300 000 for sunset and beaches in california n 350 000 for the chicago program for the design installation and maintenance of over 950 hanging baskets n 600 000 for the abraham lincoln commission n 100 000 for the police department has a population of 400 people n 2 500 000 for the space flight center for process dry cleaning capability n 500 000 for construction of the museum nand the list goes on and on n i think we could find a few places to make cuts to pay for securing our borders |
| Society, Culture | why do filipinos are using language yet there is no such a language some of them are forced to resort back to filipino language when they don't have the necessary command of further english to complete their sentence similar to should you personally go to a foreign country you'll likely have studied up on the language but no doubt you'll get into a situation where you start a sentence in the foreign language but don't have the knowledge to complete it with to english |

| Computers, Internet | can anyone tell me how to link graphics in a c and java program i want to insert graphics in a c program how can it be done please give the coding or the link where i could find the info thanks it depends which compiler u r using if u r using c c compiler then just include graphics h file in ur code and start using its functions but these graphics are simple and limited in what we want to do instead u can search on the web for comprehensive libraries there r so many libs available on web w o any cost i mean free |
| --- | --- |
| Sports | do the yellow cards in the world cup carry through to the second round they carry on to the next game if the next game is the next round then the yellow card is going to be there which causes the player to sit out |
| Sports | are wwe wrestlers are really getting hurts while fighting yes totally |
| Health | can drink water help me lose some of the bulge around my waist i was wondering if it will help me lose some weight around the middle it most certainly can i drank nearly a 2 liter bottle of water a day and lost about 30 lbs in 2 months of course it helps to diet and exercise but water nearly does the trick but careful it will also flush your system and increase your appetite |
| Politics, Government | how can a person from another country come here at the age of 65 and collect social security and not pay in no not a dime in this country i was under the impression that you must put in in order to receive this is why we are having problems with social security now i know when i turn 65 i want all of my money and interest and i don't care what they call it i have heard that too and that is crap i think you should have to pay to get it and be a citizen of the us to reap our benefits i don't know who's bright idea that was but i'm sure as soon as they let us know he will not be a very popular man |
| Politics, Government | whats a good way to raise money to get someone out of jail i need to get 1500 to get them out no i would assume you mean to make bail not pay for an escape but also remember if they make bail in most places they can not use a pubic defender since making bail shows they have or had the money to hire thier own attorney n nwork second job sell your computer tv |

| | |
|---|---|
| Education, Reference | what is number in the 50 states and what is the ranking size in place 5 residents at the last census 20th in a list of population by state n17 43 people per km ranked out of 50 na total area of 113 sq mi ranked 6th out of 50 a state on february 14 1912 state out of 50 |
| Health | lenses would you go for the hard or soft lenses does it also affect your vision if it's hard or soft go for soft contacts and the disposable kinds i knew a friend who had the hard contacts and apparently she said they were uncomfortable i've also heard that if you eyesight is really bad they use the hard contacts but if you have the choice soft |
| Health | q about is the pain all over your body or can it be just in the lower or upper please let me know the pain from can be anywhere and everywhere each person is different |
| Computers, Internet | i only got 12 free space i need get some stuff off so i can have more space so i can defrag can someone help go to n nhttp www ccleaner com |
| Education, Reference | is it no one or noone or are both correct no one is correct |
| Business, Finance | i need help starting an ebay business where do i start start at ebay they have all the info you need |
| Family, Relationships | how do i get get my boyfriend of 3 yrs to get up for work on time he always gets up 5 min before he needs to go to work we leave at 6 i get up at um let him worry about it what are you his mom |
| Education, Reference | do you have to register your homeschool in chicago illinois see org it will give you info on your state |
| Sports | do u think that usa is going to the finals thanks for the 2 points |
| Politics, Government | what does the aclu think it is doing other than being a i mean honestly free speech is important but people also have to have decency they are helping to strip the nation of our the values and that make us americans they are ensuring that no one is judged based on their actions that anything and everything goes n nthey used to protect americans right to free speech but now they are so far left they make the 9th circus court of appeals appear right wing |

| Category | Text |
|---|---|
| Politics, Government | what is a a is the holder of various important jobs including n n formerly the head priest in an when it had responsibilities n n the chief academic officer at various universities in north america n n an officer of local government including the scottish equivalent of a mayor the lord is the scottish equivalent of lord mayor in edinburgh glasgow and n n the officer in charge of military police n n sergeant a sergeant in charge of police in the british and commonwealth armies n n the administrator of a prison n n |
| Entertainment, Music | if your husband had cheap on his breath and wanted to take you in bed would you like it you mean like the mother on that movie carrie huh n nand i liked it i liked it n n |
| Entertainment, Music | how does a band register to play the 2006 sorry to tell you this but the deadline for a band to register for a at this year's festival has long passed but registration for 2007 will be available in august |
| Health | aloe vera how do you rate it oh joy bliss instant soothing totally cleared the eczema on my hand i haven't had an attack there since i started using it 8 years ago and it smells nice |
| Sports | what is what is in the nfl are you referring to a comment made about is basically doing what you want not in your gap or zone and going for the ball |
| Politics, Government | does president bush have a clue as to what is going on in iraq yes he and i wish he would take more vacations like reagan and the rest did you don't want him to make mistakes but you want him to work non stop some people are nuts not wanting to stand behind their president he's in for 3 more years why do you want him to fail that hurts us more yes thanks to our beloved troops and to the iranian guy we love iranians just not your leader he's scary |
| Sports | england currently out of form is it the players or the coach who is to be blamed english football team is now in a depression i am hardly remembering a satisfactory win for england with its new coach steve i am in a doubt whether england could atleast get a position in is it fault or the players not all the days are the days of spring every team has to suffer from a lean period which is inevitable blaming the team or the coach is not right but we should encourage the team |

| Entertainment, Music | what is this movie i saw this movie about 8 years ago and i can't remember the name of it i think the plot was something along the lines of a group of terrorists taking over a building and there is a girl who saves everybody the one line i remember in the movie is when the computer nerd says she's like bruce lee with boobs 'no with shannon mrs gene simmons |
|---|---|
| Politics, Government | so dems what now with iraq what will you do now that you have the probably both houses of congress please no answers like well whatever it is it'll be better than republicans i'm serious i would really like to know oh by the way cutting and running will terrorist organizations to think that america is weak and increase the chance that we get hit here also that will pretty much hand over a whole country for terrorists to take over fortunately most of the dems elected to congress understand that too ok so now what any real answers it would be foolish to just leave but we have to get the mechanisms in place so iraq can for themselves and perhaps get the world community to help |
| Sports | what happened to the rock in wwe rock has been taken off the roster on wwe as he is concentrating on being an action movie star and he is doing a great job i miss the rock he was such an entertaining wrestler if ya what the rock is n ni loved the peoples yeah bring back the rock i say also cheers |
| Politics, Government | why did they do an autopsy on after he died from two 500 pound bombs war against terrorist because some liberal news said it looked like he d been beaten up by our guys |
| Sports | what is royal engineer the royal engineers afc is a football team founded in under the leadership of major of the corps of royal engineers the they enjoyed a great deal of success in the winning the fa cup in n nthe cup winning side were n w lt g h sim lieutenant g c lt r m lt p g von lt c k wood lt h e lt r h stafford lt h w lt a mein and lt c n nthe team drew 1 1 against old f c with a goal from and went on to win the replay 2 0 with a goal each from and stafford n nthey have maintained their character as an amateur team as was the tradition early on in football history and have not played in top competition since the |

| Entertainment, Music | why does walmart have two versions of the same commercial the one where the lady says she went for eye drops and came back with something eye opening why is there a black and white version i dont really think its all that all type of people shop at walmart no matter what race or color what is up with that the walmart marketing team probably feel that either version will make more of an impact in different markets it is the version with the black woman plays in so called black markets the version with the white woman probably plays in all markets whether it is considered white asian latino etc n nbut fear not it's only television television is not real |
|---|---|
| Health | i think i might have a uti could someone help me out here please i have had a urinary tract infection 2 or 3 times before mind you i'm almost 15 but i've never known it before i went to the doctor for something else and i had to do a urine sample so that's why i'm not sure if i have one now or not anyway for the past 2 or 3 days i have pain when i urinate but the pain is awful immediately after the pain is near my lower pelvis and i mean the pain is bad where i have to just stand still and walk carefully because it hurts so bad does this just happen sometimes could it be because i should be starting my period soon or should i go to a doctor thank you so much for your help go to a doctor |
| Entertainment, Music | why is fresh air bad for you cause every time i am in a club i can drink as much as i want and still walk about but as soon as i go outside and the fresh air hits me thats me down on my a se sometimes unconscious why is that lmao that's what i'll blame it on i went into fresh air |
| Family, Relationships | help me with my car and commitment my husband said he'd fix our only car no later then today well he's gone to the auto store 2 times today and now he says he doesn't want to go a third time today i however need it in the early morning what can i do 2 get him to finish what he started and keep his commitment help me uh i guess find a solution on yahoo answers |
| Science, Mathematics | what are some really good sites on the elements elements as in silicon carbon etc com any element and get everything from molar mass to melting point and electron configuration ect ect |

Table 5: Visualization on Yahoo! Answers with LSTM.

