# OpenReview forum: "L-Shapley and C-Shapley: Efficient Model Interpretation for Structured Data"
_ICLR.cc/2019/Conference_

### Official Review · AnonReviewer3 · 2018-10-24
**Nice addition to Shapley literature, but could be strengthened**

**Rating:** 6
**Confidence:** 4

**Review:**

The paper proposes two approximations to the Shapley value used for generating feature scores for interpretability. Both exploit a graph structure over the features by considering only subsets of neighborhoods of features (rather than all subsets). The authors give some approximation guarantees under certain Markovian assumptions on the graph. The paper concludes with experiments on text and images.

The paper is generally well written, albeit somewhat lengthy and at times repetitive (I would also swap 2.1 and 2.2 for better early motivation). The problem is important, and exploiting graphical structure is only natural. The authors might benefit from relating to other fields where similar problems are solved (e.g., inference in graphical models). The approximation guarantees are nice, but the assumptions may be too strict. The experimental evaluation seems valid but could be easily strengthened (see comments).

Comments:

1. The coefficients in Eq. (6) could be better explained.

2. The theorems seem sound, but the Markovian assumption is rather strict, as it requires that a feature i has an S that "separates" over *all* x (in expectation). This goes against the original motivation that different examples are likely to have different explanations. When would this hold in practice?

3. While considering chains for text is valid, the authors should consider exploring other graph structures (e.g., parsing trees).

4. For Eqs. (8) and (9), I could not find the definition of Y. Is this also a random variable representing examples?

5. The authors postulate that sampling-based methods are susceptible to high variance. Showing this empirically would have strengthened their claim.

6. Can the authors empirically quantify Eqs. (8) and (9)? This might shed light as to how realistic the assumptions are.

7. In the experiments, it would have been nice to see how performance and runtime vary with increased neighborhood sizes. This would have quantified the importance of neighborhood size and robustness to hyper-parameters.

8. For the image experiments, since C-Shapley considers connected subsets, it is perhaps not surprising that Fig. 4 shows clusters for this method (and not others). Why did the authors not use superpixels as features? This would have also let them compare to LIME and L-Shapley.

---

> ### Author Response · Authors · 2018-11-15
> **Response to Reviewer 3 (Summary)**
>
> We thank the reviewer for the detailed comments and encouraging title! We have included three experiments in the updated version to address Point 5, 6,and 7 of the reviewer’s comments,  and also omit unnecessary details in the original paper. We will respond the reviewer's comments concretely below.
>
> “The paper is generally well written, somewhat lengthy and at times repetitive (I would also swap 2.1 and 2.2 for better early motivation)”
> Based on the reviewer’s request, we have shortened the paper by deleting unnecessary repetitions and details in Section 4.3 and the experiment section, and putting some of them to appendix. For example, the description of datasets is deferred to the appendix.  As a replacement, we have included a new experiment with human evaluation. On the other hand, we still keep the order of 2.1 and 2.2. The main reason is that it seems more natural to explain how importance of a feature subset is quantified first (section 2.1) before we motivate the Shapley value, which incorporating interaction based on this quantification (section 2.2).

---

> ### Author Response · Authors · 2018-11-15
> **Response to Reviewer 3 (Details)**
>
> 1. “Coefficients in Eq. (6)”
>
> The coefficients are derived from Myerson value, which can be interpreted as the Shapley value for the coalition game with a graph structure. The details can be found in the proof of Theorem 2. In particular, Equation (22) in the Appendix provides the concrete procedure of derivation.
>
> 2. "The Markovian assumption is rather strict."
>
> We thank the reviewer for addressing this point. We agree with the reviewer that Markovian assumption introduces bias in explanation, which aims for a better bias-variance trade-off when approximating Shapley values on structured data. Theorem 1 and Theorem 2 quantify the introduced bias under the setting when the Markovian assumption is approximately true. We also show on real data such an approximation achieves a better bias-variance trade-off empirically when the number of model evaluations is linear in the number of features.
>
> 3. "Use other graph structures like parse trees on language."
>
> The reviewer made a very bright proposal. As the current paper focuses on the study of the generic setting where data with graph structure, we only use the simplest possible model on language to demonstrate the validity of the proposed algorithms. But the proposed idea can be a promising future direction. The authors have been thinking along the same direction for a while. One question one could ask is whether there exists a better solution concept in coalitional game theory under the setting of a parse tree. Related literature includes [1] and [2] if the reviewer is interested to think about this further.
>
> 4. "Y in Eqs. (8) and (9)."
>
> We assume the model has the form P_m(Y|X). Y is the response variable from the model.
>
> 5. “The authors postulate that sampling-based methods are susceptible to high variance. Show this empirically.”
>
> We have added an experiment in the updated version addressing the statistical dispersion of estimates of the Shapley value produced by sampling-based methods. Two commonly used nonparametric metrics are introduced to measure the statistical dispersion between different runs of a common sampling-based method, as the number of model evaluations is varied. Figure in the link below shows the variability of SampleShapley and KernelSHAP as a function of the number of model evaluations:
> https://drive.google.com/file/d/1yUvJ_Jqn2Bg16U-poEtMcTGfWifIcQ3_/view?usp=sharing
> See also Appendix E for details.
>
> 6. "Empirically quantify Eqs. (8) and (9)."
>
> While we agree with the reviewer that a good empirical quantification of quantities in Eqs. (8) and (9) can verify the assumptions in practice, it is rather difficult to get a reliable estimate of the conditional mutual information (or similar quantities) in the high dimensional regime. We have added one experiment in the updated version to validate the correlation between our algorithms and the Shapley value directly, which partially reflects the conclusion of our theorem. See the figure in the link below and Appendix C for details: https://drive.google.com/file/d/1oWsWyA4IkDIbaOjwOOwMAYJzu6kUuQSa/view?usp=sharing
>
> The better performance on real data in terms of log-odds ratio decay when top features are masked may also be viewed as a partial empirical evidence on the fact that the introduced bias is not as big as the reduced variance.
>
> 7. “it would have been nice to see how performance and runtime vary with increased neighborhood sizes”
>
> We have included a section for sensitivity analysis of our algorithms in the updated version. We study how correlation between the proposed algorithms and the Shapley value vary with the radius of neighborhood, the only hyper-parameter in our algorithms. A plot of model evaluations against the radius of neighborhood is also included. See the figures in the link below, and also Appendix D for details:
> https://drive.google.com/open?id=1perbCh7oH95j3uDp6jNEM0vPvUvcUkZ8
> https://drive.google.com/file/d/1f5yBIwxd85tyxQKB5gBlRtBX4pRe0noL/view?usp=sharing
>
> 8. "Not use superpixels as features."
>
> We agree with the reviewer that using superpixels may lead to better visualization results. However, this leads to a performance decay in terms of the change in log-odds ratio when a fixed number of pixels are masked. The same issue has been addressed in [3]. For fairness of comparison, we use the raw pixels as features for all methods.
>
> [1] Winter, Eyal. "A value for cooperative games with levels structure of cooperation." International Journal of Game Theory 18.2 (1989): 227-240.
> [2] Faigle, Ulrich, and Walter Kern. "The Shapley value for cooperative games under precedence constraints." International Journal of Game Theory 21.3 (1992): 249-266.
> [3] Lundberg, Scott M., and Su-In Lee. "A unified approach to interpreting model predictions." Advances in Neural Information Processing Systems. 2017.

---

### Official Review · AnonReviewer1 · 2018-11-02
**Novel methods for Shapley value estimation seem theoretically sound, could benefit from slightly more extensive evaluation**

**Rating:** 7
**Confidence:** 2

**Review:**

This paper provides new methods for estimating Shapley values for feature importance that include notions of locality and connectedness. The methods proposed here could be very useful for model explainability purposes, specifically in the model-agnostic case.  The results seem promising, and it seems like a reasonable and theoretically sound methodology.  In addition to the theoretical properties of the proposed algorithms, they do show a few quantitative and qualitative improvements over other black-box methods.  They might strengthen their paper with a more thorough quantitative evaluation.

I think the KernelSHAP paper you compare against (Lundberg & Lee 2017) does more quantitative evaluation than what’s presented here, including human judgement comparisons.  Is there a way to compare against KernelSHAP using the same evaluation methods from the original paper?

Also, you mention throughout the paper that the L-shapley and C-shapley methods can easily complement other sampling/regression-based methods.  It's a little ambiguous to me whether this was actually something you tried in your experiments or not.  Can you please clarify?

---

> ### Author Response · Authors · 2018-11-15
> **Response to Reviewer 1**
>
> We thank the reviewer for the detailed suggestions and encouraging comments! We have included an experiment with human evaluation in the updated version. Below we respond to Reviewer 1’s questions in details.
>
> “Is there a way to compare against KernelSHAP using the same (human) evaluation methods from the original paper?”
>
> We agree with the reviewer that human evaluation is important in this area, and we have added a new experiment with human evaluation in the updated version.
>
> In KernelSHAP paper, the authors designed experiments to argue for the use of Shapley value instead of LIME, which shows Shapley value is more consistent with human intuition on a data set with only a few number of features. Both KernelSHAP and our algorithms are ways of approximating Shapley value when there is a large number of features, under which case the exact same experiment is difficult to replicate.
>
> We have designed two experiments by ourselves involving human evaluation for our methods and KernelSHAP on IMDB in the updated version. We assume that the key words contain an attitude toward a movie and can be used to infer the sentiment of a review. In the first experiment, we ask humans to infer the sentiment of a review within a range of -2 to 2, given the key words selected by different model interpretation approaches. Second, we also ask humans to infer the sentiment of a review with top words being masked, where words are masked until the predicted class gets a probability score of 0.1. In both experiments, we evaluate the consistency with truth, the agreement between humans on a single review by standard deviation, and the confidence of their decision via the absolute value of the score. We observe L-Shapley and C-Shapley take the lead respectively in two experiments. See the table and an example interface in the links below, and also Section 5.3 for more details:
> https://drive.google.com/open?id=1aHZPP0ZAdyODgTEFLRrQAKyS4uJ8h-XS
> https://drive.google.com/file/d/1_HOR28DGlKqEQVplGahv47o2xPe5lT5e/view?usp=sharing
>
> “It's a little ambiguous to me whether you tried to complement other sampling/regression-based methods in your experiments or not. Can you please clarify?”
>
> In the experiments, we didn't combine our approach with sampling based methods as the number of model evaluations is already small enough in the setting (linear in the number of features).

---

### Official Review · AnonReviewer2 · 2018-11-04
**A new method for computing Shapely values**

**Rating:** 7
**Confidence:** 3

**Review:**

This paper proposes two methods for instance-wise feature importance scoring, which is the task of ranking the importance of each feature in a particular example (in contrast to class-wise or overall feature importance).  The approach uses Shapely values, which are a principled way of measuring the contribution of a feature, and have been previously used in feature importance ranking.

The difficulty with Shapely values is they are extremely (exponentially) expensive to compute, and the contribution of this paper is to provide two efficient methods of computing approximate Shapely values when there is a known structure (a graph) relating the features to each other.

The paper first introduces the L(ocal)-Shapely value, which arises by restricting the Shapely value to a neighbourhood of the feature of interest.  The L-Shapely value is still expensive to compute for large neighbourhoods, but can be tractable for small neighbourhoods.

The second approximation is the C(onnected)-Shapely value, which further restricts the L-Shapely computation to only consider connected subgraphs of local neighbourhoods.  The justification for restricting to connected neighbourhoods is given through a connection to the Myerson value, which is somewhat obscure to me, since I am not familiar with the relevant literature.  Nonetheless, it is clear that for the graphs of interest in this paper (chains and lattices) restricting to connected neighbourhoods is a substantial savings.

I have understood the scores presented in Figures 2 and 3 as follows:

For each feature of each example, rank the features according to importance, using the plugin estimate for P(Y|X_S) where needed.
For each "percent of features masked" compute log(P(y_true | x_{S\top features})) - log(P(y_true | x)) using the plugin estimate, and average these values over the dataset.

Based on this understanding the results are quite good.  The approximate Shapely values do a much better job than their competitors of identifying highly relevant features based on this measure.  The qualitative results are also quite compelling, especially on images where C-Shapely tends to select contiguous regions which is intuitively correct behavior.

Comparing the different methods in Figure 4, there is quite some variability in the features selected by using different estimators of Shapley values.  I wonder is there some way to attack the problem of distinguishing when a feature is ranked highly when its (exact) Shapley value is high versus when it is ranked highly as an artifact of the estimator?

---

> ### Author Response · Authors · 2018-11-15
> **Response to Reviewer 2**
>
> We thank the reviewer for the detailed and encouraging comments! Based on the suggestions from the reviewer, we have included an experiment in the updated version that measures the correlation between L-Shapley, C-Shapley and the Shapley value.
>
> “Understanding of the evaluation metric”:
>
> The evaluation metric we use is the following:
> log(P(y_pred | x)) - log(P(y_pred | x_{top features MASKED})). The reviewer's understanding is in general correct except that we use the predicted label instead of the true label in the data set, because we hope to find key features for why the model makes its own decision.
>
> “I wonder is there some way to attack the problem of distinguishing when a feature is ranked highly when its (exact) Shapley value is high versus when it is ranked highly as an artifact of the estimator?”
>
> We have added a new experiment in the updated version to address the problem of how the rank of features correlates with the rank produced by the true Shapley value. We sample a subset of test data from Yahoo! Answers with 9-12 words, so that the underlying Shapley scores can be accurately computed. We employ two common metrics, Kendall's Tau and Spearman's Rho to measure the similarity (correlation) between two ranks. We have observed a high rank correlation between our algorithms and the Shapley value. See the figure in the link below, and also Appendix C for more details:
> https://drive.google.com/open?id=1oWsWyA4IkDIbaOjwOOwMAYJzu6kUuQSa

---

### Public Comment · (anonymous) · 2018-09-30
**Very weak baselines**

I just wanted emphasize that the baselines used in this paper are very weak.  To the best of my knowledge, no one has claimed that any of the provided baselines (LIME, KernelSHAP, or SampleSHAP) are remotely close to SOTA for, or even capable of, interpreting neural networks in the manner demonstrated here, as the original papers focused on simpler models, such as SVM, or image models with superpixel preprocessing.

The authors (partially) address this in the results section:

"We emphasize that our focus is model-agnostic interpretation, and we omit the comparison with interpretation methods requiring additional assumptions or specific to a certain class models, like Integrated Gradients (Sundararajan et al., 2017), DeepLIFT (Shrikumar et al., 2017), LRP (Bach et al., 2015) and LSTM-specific methods (Karpathy et al., 2015; Strobelt et al., 2018; Murdoch & Szlam, 2017)."

Even if limited to model-agnostic interpretation, a very simple, strong baseline is leave one out - black out a variable and see how much the prediction changes - which is well established in both NLP (https://arxiv.org/pdf/1612.08220.pdf) and vision (https://arxiv.org/abs/1311.2901). This method would perform significantly better than the provided baselines (the baseline examples in the bottom two rows of Figure 4 are the worst I've seen in any paper).

I'd also argue that gradient-based methods should be compared against, such as gradient times input or integrated gradients. While not truly model-agnostic, they only require the model to be differentiable, thus apply to all neural nets, and all models considered in this paper.

Moreover, even if not directly comparable, I'd argue that at least some model-specific techniques should be included as well, in order to see how much is lost by moving from a custom method to a model-agnostic one.

---

> ### Author Response · Authors · 2018-10-07
> **Response: Baselines not weak; Model-specific comparison not necessary in paper, but available in the reply**
>
> We first thank the reader for reading and greatly appreciate his/her time for writing such detailed reviews:)
>
> In summary, the reader proposes two suggestions:
> 1. The current baselines, including KernelSHAP and LIME, are weak, compared to methods like 'leave-one-out'.
> 2. The authors should compare with model-specific techniques, including ‘integrated gradients’.
>
> The short reply is:
> 1. Leave-one-out is not as strong as KernelSHAP, both theoretically and experimentally.
> 2. We do not compare with model-specific approaches in the paper as we focus on model agnostic interpretation. See the anonymous link at the end for a comparison made specifically for the reader.
>
> Below are the concrete details:
>
> We have different opinions on the first point (to a certain extent). In particular, KernelSHAP is stronger than 'leave-one-out':
> a. Based on the source code of KernelSHAP (https://github.com/slundberg/shap/blob/master/shap/explainers/kernel.py), KernelSHAP considers 'masking each word' when computing importance scores, as long as the number of samples is super-linear in the number of features.
> b. Shapley value further incorporates the interaction between features when the number of samples is larger than d (the number of features), which is not the case for leave-one-out.
> c. Experimentally, Leave-one-out is not as good as KernelSHAP when more than one features are masked in terms of the decay in log likelihood.
>
> Secondly, the focus of this work is on model-agnostic interpretation, and thus we did not include comparison with model-specific methods in the paper. Model-specific methods can have superior performance in some cases while suffer a performance decay in other cases: For example, Integrated Gradients can have comparable performance to L-Shapley on CNNs, but perform not as well as other methods on LSTM with comparable complexity. Comparing our methods with all model-specific methods for various models will be an unnecessary use of time and also distract readers from the focus of the paper:  efficient approximations of Shapley value, as a model-agnostic method for model interpretation. Being MODEL-AGNOSTIC can be important in some practical settings where models are not specified or multiple models are used.
>
> Nevertheless, it does no harm to compare one or two model-specific methods in the reply as suggested by the reader. The reader proposes to compare our methods with Gradient X Input, DeepLIFT and Integrated Gradients. Given the inferior performance of Gradient X Input and the complexity of implementing DeepLIFT, we only compare with Integrated Gradient on NLP tasks, where the time complexity of integrated gradients is controlled to be (approximately) the same as L-Shapley for each sample:
> https://drive.google.com/file/d/1UYp2lKDXt-ORgL5vKsU35K5SMa-GQSrs/view?usp=sharing

---

### Author Response · Authors · 2018-11-15
**UPDATED: Four New Experiments Based on the Suggestions of the Reviewers**

We have added four experiments in the updated version of the paper based on the suggestions of three reviewers. The first experiment compares human evaluation on top words selected by our algorithms and KernelSHAP, and also compares human evaluation on masked reviews. The second experiment evaluates how the rank produced by our algorithms correlates with the rank of the Shapley value. The third experiment evaluates the sensitivity of our algorithms to the size of neighborhood. The last experiment empirically evaluates the statistical dispersion of sampling-based algorithms. The first experiment has been added to Section 5.3 in the main paper while the rest are added to Appendix C,D and E.

There are also some other minor changes addressing the concern of Reviewer 3 in the length of the paper. We have deferred the detailed description of data sets and models into the appendix. We have shortened Section 4.3 which describes the connection with related work. We also reduced the number of text examples for visualization in the appendix.

We again express our sincere thanks to all the reviewers, who have helped build our manuscript into a better and more complete shape!

---

### Meta-Review · Area_Chair1 · 2018-12-14
**Solid technical novelty with convincing empirical results.**

**Confidence:** 4
**Recommendation:** Accept (Poster)

**Metareview:**

The paper presents two new methods for model-agnostic interpretation of instance-wise feature importance.

Pros:
Unlike previous approaches based on the Shapley value, which had an exponential complexity in the number of features, the proposed methods have a linear-complexity when the data have a graph structure, which allows approximation based on graph-structured factorization. The proposed methods present solid technical novelty to study the important challenge of instance-wise, model-agnostic, linear-complexity interpretation of features.

Cons:
All reviewers wanted to see more extensive experimental results. Authors responded with most experiments requested. One issue raised by R3 was the need for comparing the proposed model-agnostic methods to existing model-specific methods. The proposed linear-complexity algorithm relies on the markov assumption, which some reviewers commented to be a potentially invalid assumption to make, but this does not seem to be a deal breaker since it is a relatively common assumption to make when deriving a polynomial-complexity approximation algorithm. Overall, the rebuttal addressed the reviewers' concerns well enough, leading to increased scores.

Verdict:
Accept. Solid technical novelty with convincing empirical results.